

**Linking climatic-driven iron toxicity and water stress to a massive mangrove dieback**
James Z. Sippo[1,2], Isaac R. Santos[2,3], Christian J. Sanders[2], Patricia Gadd[4], Quan Hua[4], Catherine Lovelock[5],
Nadia S. Santini[6,7], Scott G. Johnston[1], Yota Harada[8], Gloria Reithmeir[1], Damien T. Maher[1,9]
[1]Southern Cross Geoscience, Southern Cross University, Lismore, 2480 Australia.
[2]National Marine Science Centre, Southern Cross University, PO Box 4321, Coffs Harbour,
NSW 2450, Australia
[3]Department of Marine Sciences, University of Gothenburg, Sweden
[4]Australian Nuclear Science and Technology Organisation (ANSTO), Locked Bag 2001,
Kirrawee DC, NSW 2232, Australia
[5]School of Biological Sciences, the University of Queensland, St Lucia QLD 4072, Australia
[6]Cátedra Consejo Nacional de Ciencia y Tecnología, Av. Insurgentes Sur 1582, Crédito
Constructor, Benito Juárez, 03940, Ciudad de México, Mexico.
[7]Instituto de Ecología, Universidad Nacional Autónoma de México, Ciudad Universitaria,
04500, Ciudad de México, Mexico.
[8]Australian Rivers Institute – Coast and Estuaries, and School of Environment and Science,
Griffith University, Gold Coast, QLD 4222, Australia
[9]School of Environment, Science and Engineering, Southern Cross University, Lismore 2480,
Australia
**Abstract**
*A massive mangrove dieback event occurred in 2015/2016 along ~1000km of pristine*
*coastline in the Gulf of Carpentaria, Australia. To gain insights into dieback drivers, we*
*combine sediment and wood chronologies to analyze geochemical and climatic changes. The*
*unique combination of low rainfall and low sea level observed during the dieback event was*
*unprecedented in the previous three decades. Multiple lines of evidence from iron (Fe)*
*chronologies in wood and sediment, wood densities and mangrove water use efficiency*
*suggest low water availability within the dead mangrove forest. Wood and sediment*
*chronologies suggest a rapid and large mobilization of sedimentary Fe, which was likely*
*associated with pyrite oxidation within mangrove sediments. High resolution elemental*
*analysis of wood cross sections revealed 30-90 fold increase in Fe concentrations in dead*
*mangrove areas just prior to mortality. Fe concentrations in wood samples correlated*
*strongly with the El Niño Southern Oscillation (ENSO) index, suggesting ENSO was a major*
*driver of Fe mobilization. Large Fe losses from sediments during the dieback are consistent*
*with Fe uptake in the trees, further implying sediment pyrite oxidation. If our data are*
*representative of the entire dieback region, we estimate that the dieback drove the*
*mobilization and loss of 50 ± 173 Gg Fe, equivalent to 8-50% of annual global atmospheric*
*Fe deposition into the oceans, which is one of the major drivers of surface ocean*
*productivity. Overall, our observations support the hypothesis that the forest dieback was*
*associated with low water availability and Fe toxicity driven by a strong ENSO event.*




## Introduction

Mangroves provide a wide range of ecosystem services such as nursery habitat, carbon sequestration, and coastal protection (Barbier et al. 2011, Donato et al. 2011). Climate change is a major threat to mangroves adding to the threats imposed by deforestation and over exploitation (Hamilton and Casey 2016, Richards and Friess 2016). Rising sea levels, changing sediment budgets, reduced water availability and increasing climatic extremes are negatively affecting mangroves (Gilman et al. 2008, Alongi 2015, Lovelock et al. 2015, Sippo et al. 2018). In Australia, an extensive mangrove dieback event occurred in the Gulf of Carpentaria, in December 2015 - January 2016, coincident with extreme drought and low sea level. This extreme climatic event drove the largest recorded mangrove mortality event (~1000 km coastline, ~7400 ha) driven by natural causes (Duke et al., 2017; Harris et al., 2017; Sippo et al., 2018). Two other large scale mangrove dieback events occurred at the same time in Australia. One was reported in Exmouth, Western Australia (Lovelock et al. 2017) and another occurred in Kakadu National Park, Northern Territory, Australia (Asbridge et al. 2019).

The mangrove mortality has been attributed to low water availability associated with extreme drought. The limited rainfall and groundwater availability, and anomalously low sea level reduced tidal inundation and soil water content (Duke et al., 2017; Harris et al., 2017). A strong El Niño event resulted in the lowest recorded rainfall in the nine months preceding the mangrove dieback since 1971, and regional sea levels that were 20 cm lower than normal (Harris et al., 2017). Atmospheric moisture was also unusually low during 2015, which may influence the physiological functioning of mangrove trees (Nguyen et al. 2017). The climatic and hydrologic changes may affect both plant physiology and sediment geochemistry. Here, we explore whether low sediment water content and associated changes in sediment geochemistry may have played a role in the dieback.

In contrast to terrestrial forest soils, mangrove sediments are largely anoxic due to their water-logged nature, and high organic matter contents. Mangrove sediments also receive a supply of materials from both terrestrial environments (e.g. Fe, sediments) and oceanic water (e.g. $SO_4$) which results in distinctly different biogeochemical cycling within the sediments than terrestrial forests (Burdige 2011). As a result, mangrove sediments often accumulate bioauthigenic pyrite ($FeS_2$) which remains stable under reducing conditions (van Breemen 1988, Johnston et al. 2011). Lowering of water levels can alter sediment redox conditions and result in the oxidation of $FeS_2$, releasing acid and dissolved Fe (mostly as $Fe^{2+}$) to porewater (Burton et al. 2006, Johnston et al. 2011, Keene et al. 2014). Subsequent oxidation of $Fe^{2+}$ and precipitation of $Fe^{3+}$ (oxy)hydroxide minerals can then lead to the accumulation of highly reactive Fe in sediments. Reactive $Fe^{3+}$ minerals are in turn readily subject to reductive dissolution and (re)-formation of $Fe^{2+}$ during any switches to reducing conditions. Thus, changes in sediment redox conditions, i.e. increased oxidation and consequent reduction, in mangrove sediments that are rich in $FeS_2$ can cause a release of bioavailable $Fe^{2+}$.

Mangrove uptake of $Fe^{2+}$ may be a powerful proxy of historic sediment redox conditions, driven by short and long-term variability in sea-level. High levels of bioavailable Fe may also be associated with tree mortalities, although there is limited evidence of Fe toxicity. Fe toxicity in some mangrove species can occur at concentrations ~2 fold higher than the optimal Fe supply for maximal growth (Alongi, 2010). Since Fe is often a limiting nutrient in



ocean surface water, Fe outwelling from mangroves could have important implications for
productivity in coastal waters (Jickells and Spokes , Fung et al. 2000, Holloway et al. 2016).
An extensive saltmarsh dieback in southern United States in 2000 provides an analogue to the
mangrove dieback studied here. The saltmarsh dieback coincided with drought conditions
(McKee et al. 2004, Ogburn and Alber 2006, Alber et al. 2008). McKee et al. (2004) found
that sediments in dead saltmarsh areas had significantly higher acidification upon oxidation
than alive areas. The dieback may have been caused by reduced water availability, increased
sediment salinities and/or metal toxicity associated with soil acidification following sediment
pyrite oxidation. However, the cause of the dieback was debated and remains inconclusive
(McKee et al. 2004, Silliman et al. 2005, Alber et al. 2008). In contrast to the herbaceous salt
marsh species in the US dieback, mangroves are woody, providing opportunities for climatic
reconstructions (Verheyden et al. 2005, Brookhouse 2006). To date, the use of
dendrochronological techniques have not been used to assess changes in sediment
geochemistry in mangroves.
Here, we combine multiple wood and sediment chronology techniques to reconstruct
sediment geochemistry and assess links to climate. To evaluate the potential for mobilisation
of Fe during the dieback, we combine multiple lines of evidence including: 1) micro X-ray
fluorescence (Itrax) to analyse the elemental composition in wood and sediment cores; 2)
wood density measurements, tree growth rates and $\delta^{13}C$ isotopes to assess historic changes in
water availability (Santini et al. 2012, Santini et al. 2013, Van Der Sleen et al. 2015, Maxwell
et al. 2018); and 3) sediment profiles of $FeS_2$ concentrations to give insight into sediment
redox conditions and Fe mobilisation. We assess these parameters in areas where mangroves
died and where they survived the dieback event.

## Methods

### Study Site

This study was conducted in the South Eastern corner of the Gulf of Carpentaria, in Northern
Australia (Figure 1). The Gulf of Carpentaria is a large and shallow (< 70 m) waterbody with
an annual rainfall of 900mm per year and a semi-arid climate (Bureau of Meteorology; Duke
et al., 2017). The region has low lying topography with abundant mangroves and extensive
saltpans in the upper intertidal zone (Duke et al., 2017). The species *Avicennia marina* and
*Rhizophora stylosa* dominate the mangroves within the Gulf of Carpentaria.
Widespread dieback of the mangroves in the region was observed in 2015-2016. The dieback
predominantly affected *A. marina* which occupy the open coastlines and upper intertidal
areas (Duke et al., 2017). Although 7500 ha of mangrove suffered mortality some areas
remained relatively unaffected, providing an opportunity to compare conditions in live and
dead stands. We assessed a live and dead mangrove area 20 months after the dieback event.
The two mangrove areas were separated by the Norman River and were ~ 4 km apart (Figure
1). The living mangrove has an area of 175 ha and had some dead trees in the upper intertidal
zone and living trees that showed signs of stress (dead branches and partial defoliation).
Towards the seaward edge, the forest had no signs of canopy loss 8 months post dieback
event. The dead mangrove area was 169 ha and had close to 100% mortality (Figure 1b), with
only some trees at the waterline showing regrowth.





**Figure 1. Study sites of a) living mangrove area (green) and b) dead mangrove area (red) near to the mouth of the Norman River, Karumba Qld. Note: The yellow 'x' symbols represent transects through the upper, middle and lower study sites. c) Elevation above the Australian Height Datum (AHD) from Lidar DEM were measured from the seaward mangrove edge in 2017 from the same transects as samples were collected in 2016 through the living (Green line) and dead (red line) mangrove area (data available from http://wiki.auscover.net.au/wiki/Mangroves). Satellite images sourced from © Google Earth (2019) and Queensland Government (2019).**

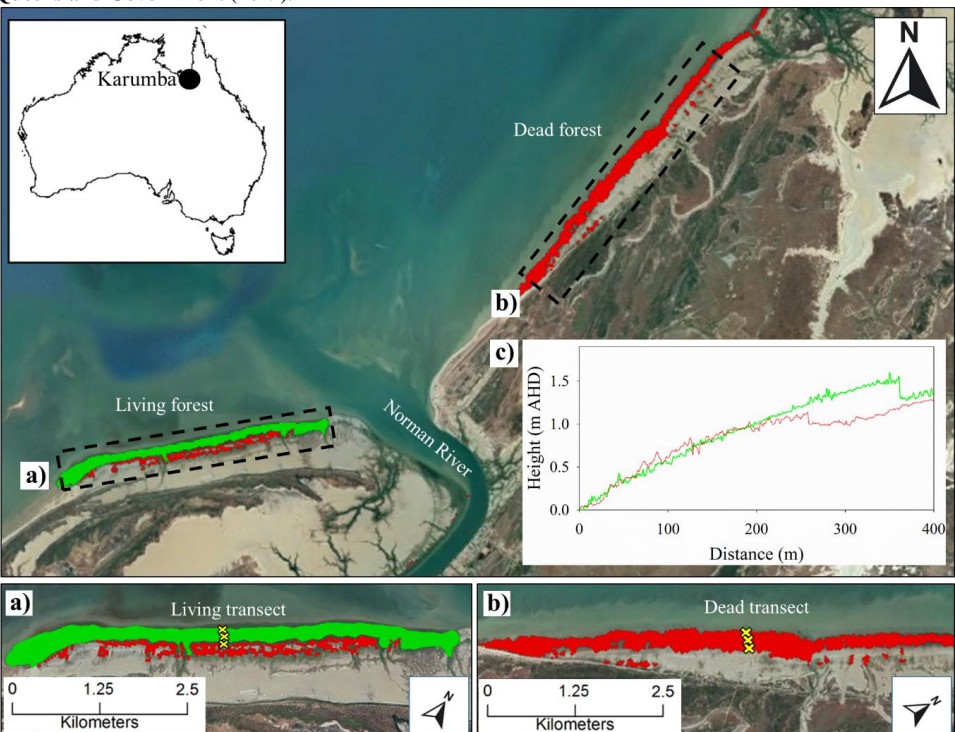

**Field sampling and chemical and isotopic analyses**

Tree and sediment samples were collected in August 2017 approximately 20 months after the dieback event. Wood and sediment samples were collected from transects from the lower intertidal zone to the upper intertidal zone (Figure 1). Fully mature trees were selected at ~20 m inward from the lower and upper intertidal forest edges and in the centre of the forest. One upper, mid and lower tide wood sample was taken in living and dead mangrove areas (Figure 1a and b). Wood samples from *A. marina* were taken from 50 cm above ground level by cutting a 1cm thick disk from the trunk. At the upper and lower intertidal sites, two sediment cores were taken. One core, taken to 2 m with a Russian peat auger with extensions, was sampled for elemental analysis with Itrax. A second core, taken to a depth of 1 m using a tapered auger corer in August 2018 at the same site, was sampled for analysis of chromium reducible sulfur (CRS).

Wood samples were dated using bomb $^{14}$C (eg, Santini et al. 2013; Witt et al. 2017). Water-use efficiency (WUE), which is the ratio of net photosynthesis to transpiration, was assessed using wood cellulose stable isotopic composition $\delta^{13}$C following (Van Der Sleen et al. 2015)



as water use efficiency correlates with larger values of $\delta^{13}C$ (Farquhar and Richards 1984,
Farquhar et al. 1989). Wood elemental composition and density was assessed using micro X-
ray fluorescence. Sub-samples for $^{14}C$ and $\delta^{13}C$ were taken along the longest radius of each
wood disk at regular intervals from the centre of the wood disc to the outer edge (youngest
wood). The sub-samples were collected using a scalpel parallel to tree rings to reduce errors.
Alpha cellulose was extracted from the wooden sub-samples (Hua et al. (2004b), combusted
to $CO_2$ and converted to graphite (Hua et al. 2001). A portion of graphite was used for the
determination of $\delta^{13}C$ for isotopic fractionation correction using a Micromass IsoPrime
elemental analyser/isotope ratio mass spectrometer (EA/IRMS) at the Australian Nuclear
Science and Technology Organisation (ANSTO). The remaining graphite was analysed for
$^{14}C$ using the STAR accelerator mass spectrometry (AMS) facility at ANSTO (Fink et al.
2004) with a typical analytical precision of better than 0.3% (2σ).
Wood samples and sediment cores were analysed for elemental composition with a micro X-
ray fluorescence conducted at ANSTO using an Itrax core scanner (Cox Analytical Systems).
The scanner produces a high resolution (0.2 mm) radiographic density pattern and semi-
quantitative elemental profiles for each sample. The Itrax measured 34 elements, but we only
report Fe. Wood samples were scanned along the same transect as for $^{14}C$ samples, i.e. the
longest radius from the wood core to the outer edge. Sediment cores were analysed using the
Itrax in four 50 cm increments. Chromium reducible sulfur (CRS) was measured from
sediment samples collected with a Russian peat auger to 1 m depth to provide an estimate of
reducible inorganic S (RIS; Burton et al. (2008). CRS subsamples were placed in bags with
air removed and frozen prior to CRS analysis. Groundwater salinity values were taken at the
same sites as wood samples from bore holes dug to ~1m depth. Groundwater in the holes was
purged and allowed to refill and salinities were measured using a Hach multi-sonde.
**Data analysis**
Radiocarbon values were used to calculate calendar ages. To align dates with Itrax data, we
interpolated ages using the wood circumference. Itrax elemental and density data were
normalized as the mean subtracted from each value divided by the standard deviation
following the calculation of Z-scores by Hevia et al. (2018) and are referred to hereforth as
relative concentrations. This normalization allows a more direct comparison between samples
from living and dead areas. Growth rates in mm per year were calculated as the measured
increment divided by the difference in years (calculated from $^{14}C$) between samples. De-
trended growth rates were then calculated as the deviation from the exponential curve fitted
to growth rates for each sample. Water use efficiency (WUE) was calculated from $\delta^{13}C$
isotope values (Van Der Sleen et al. 2015). Differences in WUE between living and dead
mangrove areas were compared using T-test.
Cross correlations with a time lag of one-month intervals were used to evaluate the
relationships between climatic variables (the Southern Oscillation Index (SOI), sea level,
rainfall and vapour pressure) with wood density, elemental relative concentrations and
growth rates. SOI data and other climate data were obtained from the Bureau of Meteorology
(Station number 029028, 2019) and published reports (Jones et al. 2009, Harris et al. 2017).
All climatic data were used with a one month resolution and were standardised using a
centred moving mean.
**Results**
**Climatic conditions**





The climate records over the last three decades reveal an unprecedented combination of low
sea level and low annual rainfall. SOI is significantly correlated to all climate variables
(Pearson product moment correlation, P < 0.05). Lower sea levels and rainfall had previously
occurred independently (Figure 2). Since 1985, trends in SOI index based on vapour pressure,
precipitation and sea level observations show El Niño in 1983, 1987, 1992, 1994, 1998, 2015
and 2016.
**Figure 2. Centred moving means of climate data from the South Eastern Gulf of Carpentaria Australian**
**(Jones et al. 2009, Harris et al. 2017, Bureau of Meteorology 2019). The grey bar represents the period**
**during which the dieback event occurred.**

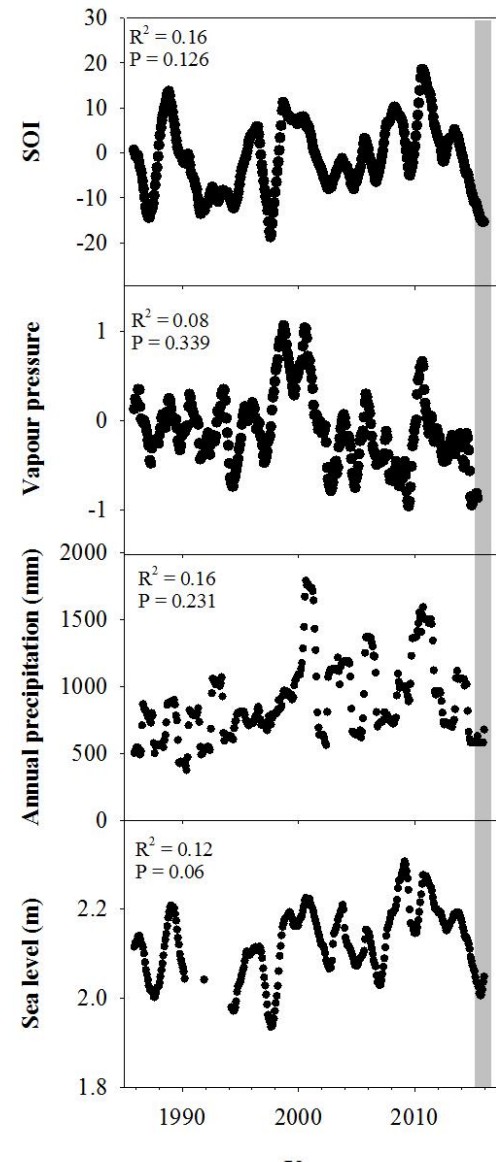




**Wood samples and ages –**The ages of *A. marina* ranged from 15 ± 2 to 34 ± 2 years (Table 1). On average, the trees in the living and dead mangrove forests were 21 ± 4 and 34 ± 1 years old respectively. Tree growth rates that were de-trended to negative exponential growth had no trends over time in either the living or dead mangrove areas (Table 1).

**Table 1. Summary of radiocarbon ages and growth rates (deviation from exponential growth) for all wood samples taken from dead and living mangrove areas in the Gulf of Carpentaria, Australia.**

| Sample | Distance from pith (mm) | $^{14}$C Mean ±2σ (pMC) ** | | | Modelled calendar age Mean ±1s (year AD) | | | Deviation from negative exponential growth (mm per year) |
|---|---|---|---|---|---|---|---|---|
| **Dead mangrove** | | | | | | | | |
| **Upper intertidal** | 2 | 122 | ± | 0.3 | 1984 | ± | 2 | - |
| | 17 | 119.8 | ± | 0.3 | 1986 | ± | 2 | -2.6 |
| | 35 | 118 | ± | 0.3 | 1988 | ± | 2 | -1.4 |
| | 52 | 116.1 | ± | 0.3 | 1990 | ± | 3 | -1.2 |
| | 70 | 110.9 | ± | 0.3 | 1998 | ± | 2 | -4.7 |
| | 87 | 105.4 | ± | 0.2 | 2010 | ± | 2 | -1.3 |
| | 89 | | | | 2015* | | | -0.9 |
| **Mid intertidal** | 2 | 123.6 | ± | 0.3 | 1983 | ± | 2 | - |
| | 12 | 122.8 | ± | 0.3 | 1984 | ± | 2 | 2.3 |
| | 24 | 119.1 | ± | 0.3 | 1987 | ± | 2 | -4.8 |
| | 36 | 115.9 | ± | 0.4 | 1991 | ± | 3 | -3.6 |
| | 49 | 110.1 | ± | 0.3 | 1999 | ± | 2 | -3.7 |
| | 62 | 105.2 | ± | 0.3 | 2011 | ± | 3 | -0.2 |
| | 64 | | | | 2015* | | | -0.2 |
| **Lower intertidal** | 2 | 123.3 | ± | 0.4 | 1983 | ± | 2 | - |
| | 23 | 120.4 | ± | 0.4 | 1986 | ± | 2 | -2.3 |
| | 45 | 117.4 | ± | 0.4 | 1989 | ± | 2 | -1.8 |
| | 89 | 110.9 | ± | 0.3 | 1998 | ± | 2 | -1.9 |
| | 110 | 105.8 | ± | 0.3 | 2009 | ± | 2 | -2.1 |
| | 113 | | | | 2015* | | | -2.5 |
| **Living mangrove** | | | | | | | | |
| **Upper intertidal** | 2 | 163.8 | ± | 0.5 | 1995 | ± | 2 | - |
| | 20 | 112 | ± | 0.4 | 1996 | ± | 3 | 2.3 |
| | 40 | 109.8 | ± | 0.4 | 2000 | ± | 3 | -0.8 |
| | 58 | 103.7 | ± | 0.4 | 2013 | ± | 2 | -2.3 |
| | 60 | | | | 2017* | | | -2.9 |
| **Mid intertidal** | 2 | 113.3 | ± | 0.5 | 1994 | ± | 2 | - |
| | 16 | 111.1 | ± | 0.3 | 1997 | ± | 2 | -1.0 |
| | 33 | 109.2 | ± | 0.4 | 2001 | ± | 2 | 0.8 |
| | 49 | 106.6 | ± | 0.3 | 2014 | ± | 2 | -1.2 |
| | 50 | | | | 2017* | | | -2.3 |





| | | | | | | | | |
|---|---|---|---|---|---|---|---|---|
| **Mid intertidal** | 2 | 113.4 | ± | 0.3 | 1993 | ± | 3 | - |
| | 25 | 110.9 | ± | 0.3 | 1998 | ± | 2 | -1.0 |
| | 50 | 101.9 | ± | 0.3 | 2017 | ± | 1 | 0.2 |
| | 51 | | | | 2017* | | | -2.3 |
| **Lower intertidal** | 2 | 108.8 | ± | 0.3 | 2002 | ± | 2 | - |
| | 17 | 107.3 | ± | 0.3 | 2005 | ± | 2 | -5.1 |
| | 33 | 104.9 | ± | 0.4 | 2011 | ± | 3 | 9.2 |
| | 46 | 104.3 | ± | 0.3 | 2014 | ± | 2 | -2.3 |
| | 48 | | | | 2017* | | | -2.2 |

* Date of collection of *A. marina* samples
**Measured $^{14}$C content is shown in percent Modern Carbon (pMC; Stuiver and Polach

224 1977)


**Fe in wood and sediment cores -** Fe relative concentrations in all dead mangrove samples
peaked at the time of mangrove mortality in late 2015/early 2016 (Figure 3). In the living
mangrove samples, Fe peaked in late 2015/early 2016 and then decreased in 2016 and 2017
back to average levels. Peak Fe concentrations in the upper, mid and lower intertidal areas of
the dead mangrove samples were 40, 90 and 30 fold higher than their mean baseline
concentrations, respectively. In the living mangrove area, peak Fe concentrations in the
upper, mid and lower intertidal areas were 25, 4 and 3 fold higher than their mean baseline
concentrations, respectively. In the dead mangrove area, Fe levels were similar from the
upper to the lower intertidal zone (Figure 3). In the living mangrove area, Fe was highest in
the upper and mid intertidal zone and decreased in the lower intertidal zone.



**Figure 3. Fe relative concentrations in mangrove wood over time in living (green dots) and dead (red**
**dots) from upper, mid and lower intertidal areas of mangroves of the Gulf of Carpentaria, Australia.**
**Grey areas indicate the dieback event.**

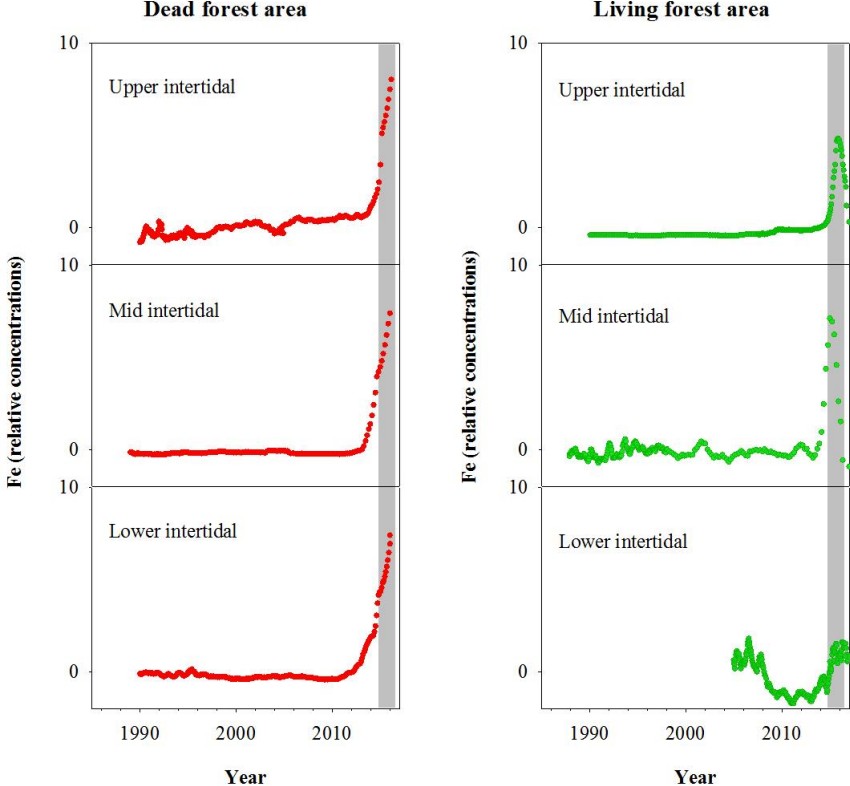



Significant correlations with no time lag were found between Fe in wood and vapour
pressure, rainfall, sea level and SOI (Figure 4). All climate variables were strongly correlated
with SOI. Therefore, we could not separate the influence of individual climate variables on
wood Fe. In the dead and living mangrove areas, the strongest correlations with Fe occurred
with no time lag (Figure 4).




**Figure 4. Cross correlation function (CCF) between Fe in wood samples and climate data at one month resolution over a 12 month period prior and post dieback. Wood samples are from the upper, mid and lower intertidal zones of the dead (red) and living (green) mangrove areas. Blue horizontal dashed lines indicate P< 0.01 with n=125. Grey dashed vertical lines at zero lag indicate dieback period and the grey bar represents the period during which the dieback event occurred.**

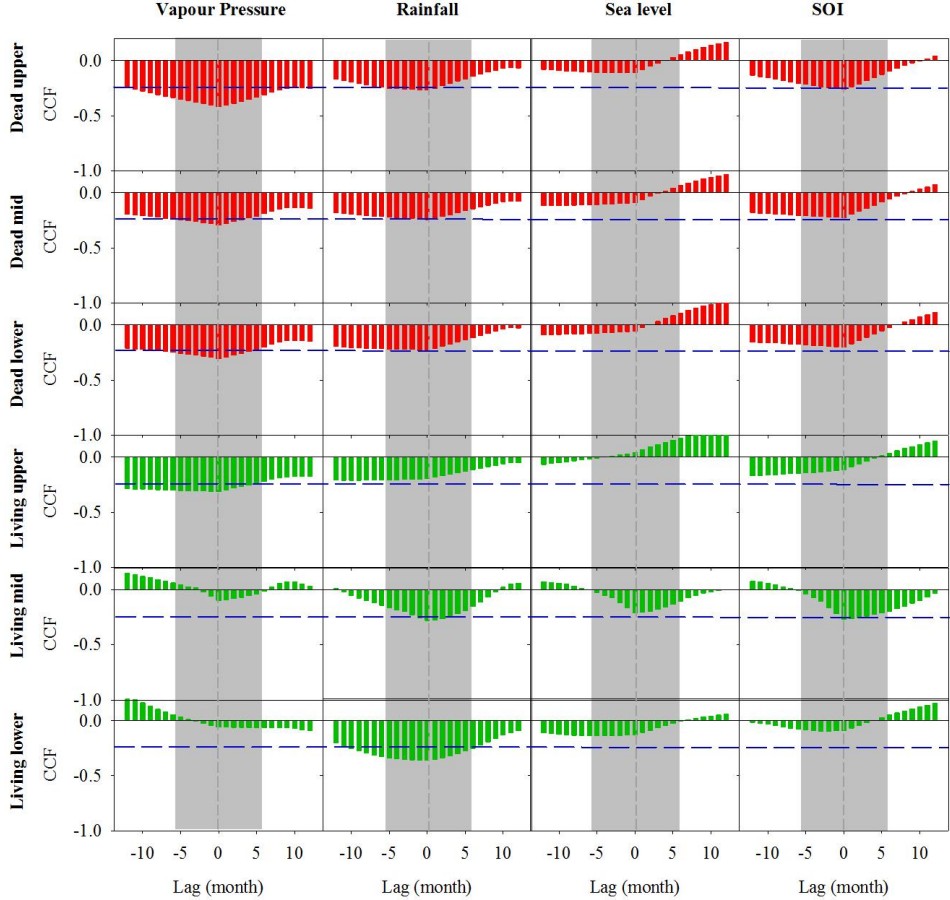

Sediment cores had a similar pattern of decreasing Fe with depth in upper and lower intertidal areas, and in living and dead mangrove areas (Figure 5a). Dead mangrove areas were depleted with Fe by ~32% in the surface 50 cm and ~26% in the surface 1 m relative to the respective living mangrove areas in both the upper and lower intertidal area (Figure 5b, c and d). Fe relative concentrations were significantly higher in living mangrove areas compared to dead mangrove areas (Mann-Whitney Rank Sum Test, $P < 0.001$ for all depths).



**Figure 5. a) Fe relative concentrations in sediment cores to 2m depth from the upper and lower intertidal**
**areas of living (green) and dead (red) mangroves in the Gulf of Carpentaria. b) Box plots of normalised**
**Fe relative concentrations from sediment cores to b) 0.5 m, c) 1 m and d) 2 m depth. The central**
**horizontal line represents the median value, the box represents the upper and lower quartiles, and the**
**whiskers represent the maximum and minimum values excluding outliers, i.e., black dots.**

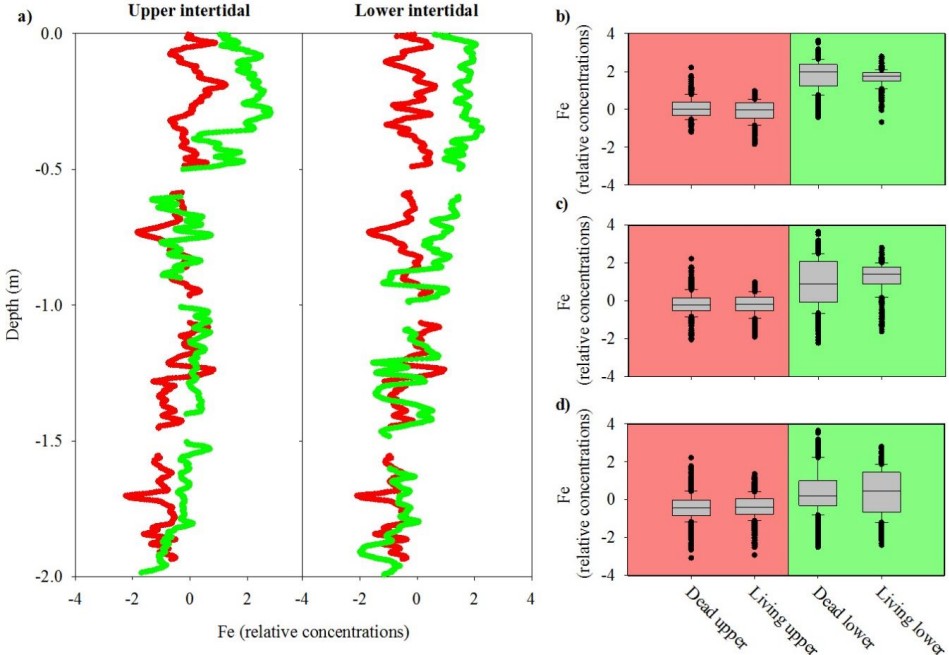


Chromium Reduced Sulfur (CRS) concentrations in sediment cores were lower in the dead
than the living mangrove area by 36% in the upper and 38% in the lower intertidal zones.
These differences were not significant (Mann-Whitney Rank Sum Test, P > 0.05) but were
very similar to Itrax differences (Figure 6). Dead mangrove sediment CRS concentrations
increased from ~10 cm depth. Living mangrove CRS increased from ~30 cm depth compared
to lower levels deeper in the soil profile, implying some recent CRS loss. In the lower
intertidal zone, CRS concentrations were highest from ~10 cm below the surface in both dead
and living sediment samples and then decreased with depth.



**Figure 6. Chromium reducible sulfur (CRS) profiles from sediment cores in dead (red) and living (green)**
**mangrove areas in the Gulf of Carpentaria.**

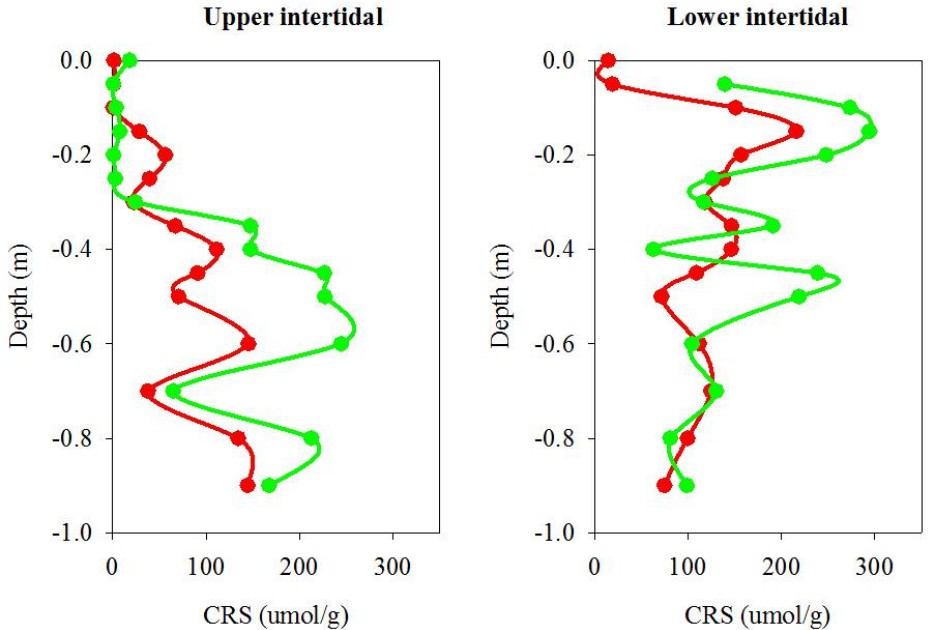


Water use efficiency (WUE) calculated from $\delta^{13}C$ decreased in all wood samples from 1983
to 2017 (Figure 7a), suggesting increasing water availability in the Gulf of Carpentaria.
During the dieback event, median WUE values were higher in dead samples than in living
samples, with the differences more pronounced in the upper intertidal zones (Figure 7b).
Comparison of WUE in dead and living mangrove samples suggests lower water availability
in the dead mangrove area (Figure 7b). However, the mean WUE values were compared from
1983 to 2017 and were not significantly different (T-test, P = 0.2) in dead and living
mangrove areas. Groundwater salinity values were highest in the upper intertidal mangrove
areas and lowest in the lower intertidal areas (Figure 7c). Salinities were not significantly
different in the living and dead forest areas (T-test, P = 0.913).





Figure 7. a) Changes to Water Use Efficiency (WUE) over time in wood samples collected from the
upper, lower and mid intertidal zone in living (green) and dead (red) mangrove areas. The grey bar
represents the mangrove dieback event. Error bars represent standard error but are not visible due to
low error of individual samples. b) Box plot of water use efficiency in mangrove wood samples in dead
and living mangrove areas in the upper, mid and lower intertidal zones. Sample size > 4 from each wood
sample. The central horizontal line represents the median, the box represents the upper and lower
quartiles, and the whiskers represent the maximum and minimum values. c) Box plot of groundwater
salinity eight months post dieback event in dead and living mangrove areas in the upper, mid and lower
intertidal zones. Sample size > 3 from each intertidal zone.

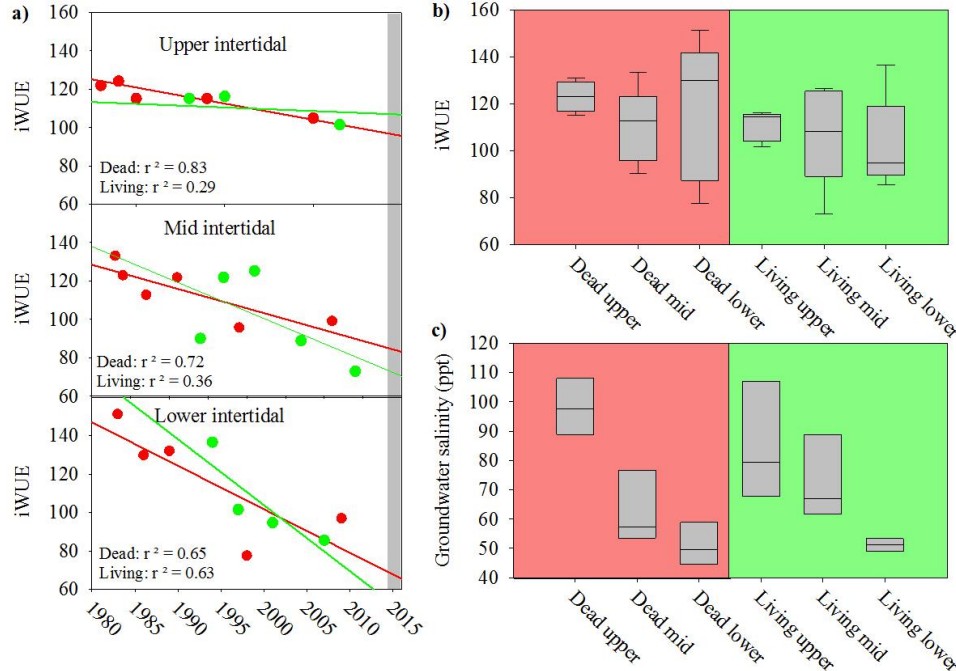






Normalised wood density values in the dead mangrove forest showed no change during the
dieback event in the upper intertidal zone, but a decline in density values occurred in the mid
and lower intertidal zones (Figure 8). In the living mangrove area, declines in wood density
values occurred in the upper and mid intertidal zones during the mortality event, but no
variation in density occurred in the lower intertidal zone (Figure 8).
**Figure 8. Normalised wood density (relative concentrations) in mangrove wood over time in living (green**
**dots) and dead (red dots) mangrove areas of the Gulf of Carpentaria, Australia. The grey bar represents**
**the time period of the dieback event.**

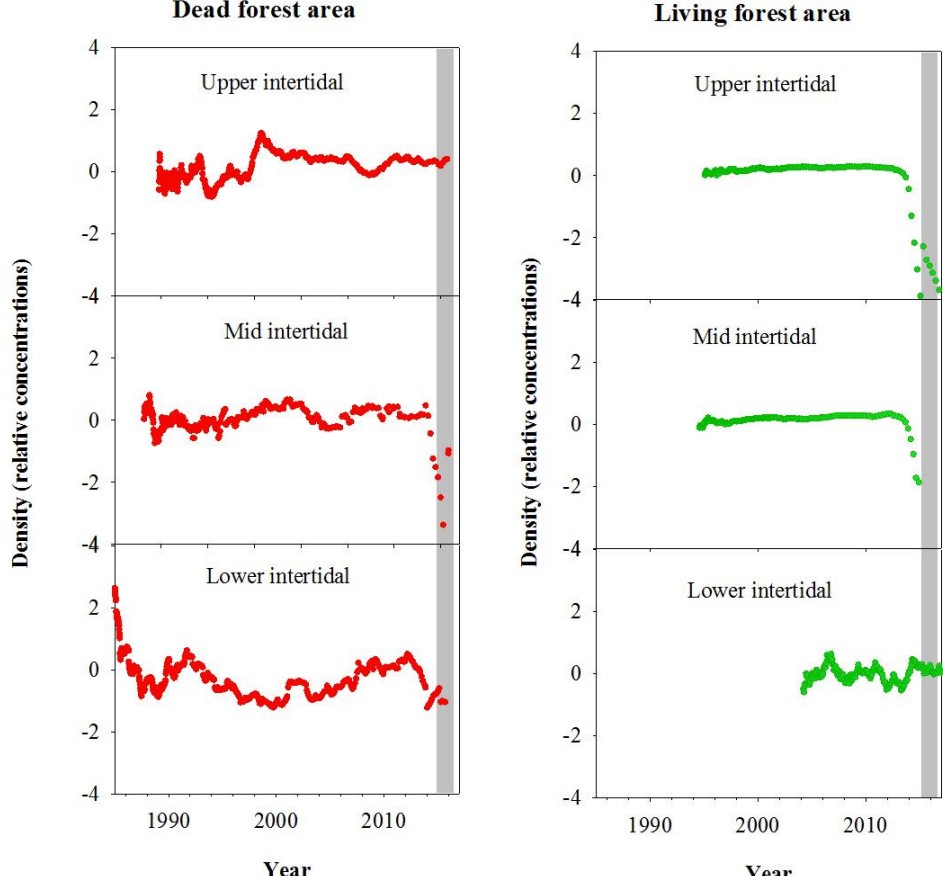




**Discussion**

**Evidence of changes in water availability from dendrogeochemistry**

Multiple lines of evidence from wood samples and sediment cores demonstrated differences in water availability between the dead and living mangrove areas. Most importantly, Fe trends in wood and sediment samples suggest the mobilisation of bioavailable Fe as $Fe^{2+}$ when bioauthigenic pyrite is oxidised and is also remobilised during the reduction of $Fe^{3+}$ if conditions return to anaerobic (Johnson et al. 2011). The most likely cause for a shift from reducing to oxidising conditions in the sediment is a reduction in water content (Keene et al. 2014) associated with the intense El Niño of 2015/16 and associated low sea level and annual rainfall (Figure 2). Trends in wood density, mangrove growth rate and water use efficiency also reveal distinct differences in water availability between dead and living forest areas.

**Fe in wood -** Elemental composition from wood samples provide strong evidence that the mangrove forest experienced sharp changes in sediment geochemistry during the dieback phase (Figure 3). This is consistent with low sea level and low rainfall/groundwater reducing soil water content, leading to oxidation of Fe sulphide minerals and release of $Fe^{2+}$. The Fe peak in the dead mangrove area at the time of tree mortality were 30 to 90 fold higher than baseline Fe (the mean Fe concentration in the sample prior to the dieback event). In the living mangrove area, an Fe peak 25 fold higher than baseline Fe was observed in the upper intertidal zone (Figure 1). In the mid and lower intertidal areas of the living mangroves, Fe peaks were 4 and 3 fold higher than baseline respectively. In all living wood samples, Fe decreased after the dieback event, which suggests that Fe in new wood growth was reduced in associated with a return to reducing sediment conditions and a concomitant reduction in $Fe^{2+}$ availability.

Records of all climate variables are in resolution of months, but the chronology of Fe (based on [14]C dates) is in years. We therefore used time lag analysis to examine relationships between climate variables and Fe over a two year period (Figure 4). Fe wood concentrations over time were significantly correlated with rainfall and vapour pressure in the dead and living forest areas (Figure 4). However, because all climate variables were strongly correlated to each other, we were unable to separate the relationships between individual climate drivers and Fe trends. We anticipate that the combination of low availability of fresh groundwater and low sea level associated with the strong El Niño event of 2015/16 are drivers of the sediment redox conditions which are reflected in wood Fe trends.

Considering the extreme increases in Fe concentrations observed in the wood samples coinciding with the dieback event, Fe toxicity could have caused mangrove mortality. Alongi, (2010) found that Fe toxicity occurred in some mangrove species at high concentrations (100mmol m$^2$ d$^{-1}$ of water-soluble Fe-EDTA) that were approximately 2 fold higher than the Fe supply for maximal growth. However, *A. marina* (the dominant species affected by the dieback at the study site) were unaffected by Fe at these concentrations. Fe concentrations can be high in acid sulphate soils, but Johnston et al. (2016) observed no *A. marina* mortality in Fe concentrations of 7-15 fold above normal. To our knowledge, no research has tested the toxicity of Fe in *A. marina* at highly elevated concentrations of bioavailable $Fe^{2+}$.
Considering that other mangrove species are affected by Fe toxicity at twofold the optimal Fe availability, it is quite possible that a 30-90 fold increase in Fe could be an additional stressor to mangroves already stressed by low water availability.



**Fe in sediments -** Sediment cores also had differences in down core Fe profiles between
living and dead mangrove areas (Figure 5a and b). Normalized Fe concentrations were lower
in the upper 1 m of sediments in the dead mangrove area compared to the living but were
very similar in sediments deeper than 1 m (Figure 5a and b). Similar trends were also
observed in CRS sediment core profiles, which have ~40 % lower $FeS_2$ concentrations in the
dead mangroves in comparison to the living mangrove sediments (Figure 6). Although
mangrove sediment conditions are highly heterogenous (Zhu et al. 2006, Zhu and Aller
2012), the sediment core results are consistent with the wood data. Both sediment and wood
suggest a general trend of iron sulphide mineral oxidation, leading to increased Fe mobility in
the dead mangrove area compared to the living area.

Sediment Fe losses due to variable periods of oxidation and reduction, as implied by Fe
profiles (Figures 3, 4 & 6), would also suggest an outwelling of Fe to the ocean. We estimate
Fe outwelling by comparing $FeS_2$ concentrations in living and dead mangrove sediment cores
based on the assumptions that (1) all Fe was originally in the form of $FeS_2$ and (2) tree Fe
uptake is a minor loss pathway. The losses of Fe from the dead mangrove sediment would be
equivalent to $87\pm163$ mmol $m^2$ $d^{-1}$. The replication of CRS sediment cores (n = 4) limits the
accuracy of our estimates. However, these fluxes are remarkably similar to short-term
porewater-derived dissolved Fe fluxes ($79\pm75$ mmol $m^2$ $d^{-1}$) estimated for a healthy temperate
saltmarsh/mangrove system (Holloway et al. 2018), building confidence in our estimates.

If our sediment cores in dead and living mangroves are representative of changes within the
entire dieback area (7400 ha), then total Fe losses from the dieback event could be equivalent
to $87\pm163$ Gg Fe. This loss is equivalent to 12 – 50% of global annual Fe inputs to the
surface ocean from aerosols (Jickells et al. , Fung et al. 2000, Elrod et al. 2004). Since the
surface ocean can be Fe limited, the consequences of Fe outwelling from this dieback event
of such a magnitude could have an intense influence on ocean productivity.

**Wood density, growth trends and water use efficiency –** Trends in wood density also
suggest differences in water availability between the living and dead mangrove areas. Clear
decreases in normalised wood density were observed during the mangrove mortality event
(Figure 8). Similar to trends in wood Fe, the wood density values in the living and dead forest
areas were correlated to climatic indicators (Appendix 1). Unlike most trees, *A. marina* has
decreased wood density with decreased growth rates due to a reduction in large xylem vessels
(Santini et al., 2012). Therefor the observed decreases in wood density likely reflect lower
growth rates, however the annual scale resolution of $^{14}$C ages prevented detection of this
short term change in growth rate. These clear decreases in wood density prior to tree
mortality are therefore an indication of stress since decreased growth rates of mangroves can
be associated with decreased water availability (Verheyden et al. 2005, Schmitz et al. 2006,
Santini et al. 2013) which is also directly related to increased salinity. Low rainfall conditions
and increased temperatures increase both evaporation and evapotranspiration while reducing
freshwater inputs (Medina and Francisco 1997, Hoppe-Speer et al. 2013), partially explaining
tree stress prior to the dieback.

Interestingly, no decrease in density was observed in the upper intertidal area of the dead
mangroves (Figure 8), despite the clear increase in Fe during the dieback in this tree (Figure
3). This suggests that no change in growth occurred prior to tree mortality, implying rapid
mortality in this case. The upper intertidal area of the dead mangroves may have been living
at the limit of its tolerance range for water availability or salinity prior to the dieback, as
suggested by extremely high groundwater salinities in the intertidal areas of dead and living



mangrove forests eight months post dieback event (Figure 7c). No decrease in wood density
was observed in the lower intertidal area of the living mangroves, which is consistent with
both variation in concentration of Fe and tree growth rate data. Together, these data suggest
that the lower intertidal area of the living mangroves was not exposed to the same conditions
during the dieback event as areas in the dead mangroves higher in the intertidal zone (Figures
3, 8 & 9). These results suggest a gradient of water availability, from extremely low
availability at the upper intertidal zone of the dead mangrove area to high/optimal availability
at the lower intertidal zone of the living mangrove area. Since the elevation profiles are
similar in the dead and living mangrove areas in the lower and mid intertidal areas (Figure 1),
it is possible that the difference in Fe trends between the mangrove areas are associated with
groundwater flows.
Mean growth rates of trees in living ($4.4 \pm 3.6$ mm yr$^{-1}$) and dead ($5.3 \pm 3.5$ mm yr$^{-1}$)
mangrove areas are similar to rates measured by Santini et al., (2013) in *A. marina* in arid
Western Australia (4.1 to 5.3 mm yr$^{-1}$). However, there was ~10 fold greater variability
because samples were collected from the upper, mid and lower intertidal zone, while Santini
et al. (2012) sampled from the lower intertidal zone only. De-trended growth rate data
showed no consistent differences in growth trends were identified between mangrove areas
(Figure 8). The lower intertidal sample of the living mangroves grew more quickly during the
dieback, which may suggest optimal conditions during this time. This may be due to
increased nutrient availability due to litterfall inputs of organic matter from nearby stressed
trees. All other samples show no indication of reduced growth prior to or during the mortality
event (Table 1). We suggest that climatic conditions drove very low growth rates during the
dieback event, as indicated by wood density data (Figure 8) and previous studies that found
low growth during droughts (Cook et al., 1977; Santini et al., 2013).
A significant difference in mean $\delta^{13}$C and WUE between living or dead mangrove areas was
observed in the upper intertidal zone (T-test, $P = 0.02$), but not in the mid or lower intertidal
zones (Figure 7b). This is consistent with the zonation of mangrove mortality which occurred
predominantly in the upper intertidal areas (Duke et al., 2017). The consistent decrease in
WUE suggests that water availability has been increasing over time in all intertidal areas
since the 1990's (Figure 7a). This is supported by generally increasing precipitation since
1980's (Figure 2), which enhanced mangrove areas in the Gulf of Carpentaria prior to this
dieback event (Asbridge et al. 2016). Therefore, climatic conditions were initially favourable
over the plants lifetime and trees may have been insufficiently acclimated to withstand
drought and low soil water availability during the dieback. Overall, this highlights the
important role of extreme climatic events counterbalancing mangrove responses to gradual
climate trends (Harris et al. 2018).
**Limitations** - This study is inherently limited in its spatial extent. Thus, the differences in Fe
between samples from living and dead mangrove areas may be due to factors other than
mangrove mortality. However, the consistency of results from multiple methods and sample
types gives confidence in the interpretation that recent changes in sediment geochemistry
have occurred associated with extreme drought and low sea level events.
Our analysis benefited from the development of high precision $^{14}$C dating of mangrove wood
samples (with age uncertainties of 1-3 calendar years; see Table 1) that rely on atmospheric
bomb $^{14}$C content resulting from above-ground nuclear testing mostly in the late 1950's and
early 1960's (Hua and Barbetti 2004a). The complexity in the wood development of *A.*



*marina* creates uncertainties (Robert et al. 2011). *Avicennia marina* secondary growth is
atypical, displaying consecutive bands of xylem and phloem which can result in multiple
cambia (i.e. the tissue providing undifferentiated cells for the growth of plants) being
simultaneously active (Schmitz et al., 2006; Robert et al., 2011). Furthermore, *A. marina*
cambia display non-cylindrical or asymmetrical growth (Maxwell et al. 2018). These
characteristics of *A. marina* atypical growth can influence our results as there is variation
within each stem.
As younger wood grows on the exterior of the tree, errors associated with ages do not
introduce uncertainty in the direction of trends but decrease the ability of finding correlated
trends with climatic variables (Van Der Sleen et al. 2015). In spite of these uncertainties, the
strong cross correlations displayed in Figure 4, with minimal time lag suggest that the
dendrochronology results are robust, and that climate variability drives long-term Fe cycling
in the coastal mangroves of the Gulf of Carpentaria.

**Summary and Conclusions**

Differences in sediment redox conditions during the dieback event were evident in all wood
and sediment data from living and dead mangrove areas. Patterns in Fe concentrations in
wood and sediment samples and climate, suggest that sediment oxidation occurred in
combination with unprecedented low sea levels and low rainfall. As the elevation of dead and
living mangrove areas was very similar, we suggest that the differences in tree survival
between areas were probably due to higher groundwater availability in the living site. The
increased oxygen permeation into sediments likely resulted in the oxidation of bioauthigenic
pyrite, which transformed into aqueous and bioavailable $Fe^{2+}$. If further oxidation occured,
$Fe^{2+}$ transforms into particulate Fe oxides. These Fe oxides are highly reactive and any
subsequent short-term reduction (e.g. with tidal inundation) would also result in mobilization
of Fe as $Fe^{2+}$. Evidence of plant Fe uptake and losses of Fe from sediments are consistent
with this hypothesized Fe mobilization. The dieback event was likely a period of transitioning
redox states in a heterogenous sediment matrix, which resulted in areas of mangrove
sediments with low water availability combined with porewaters enriched in bioavailable Fe
(Figure 9).



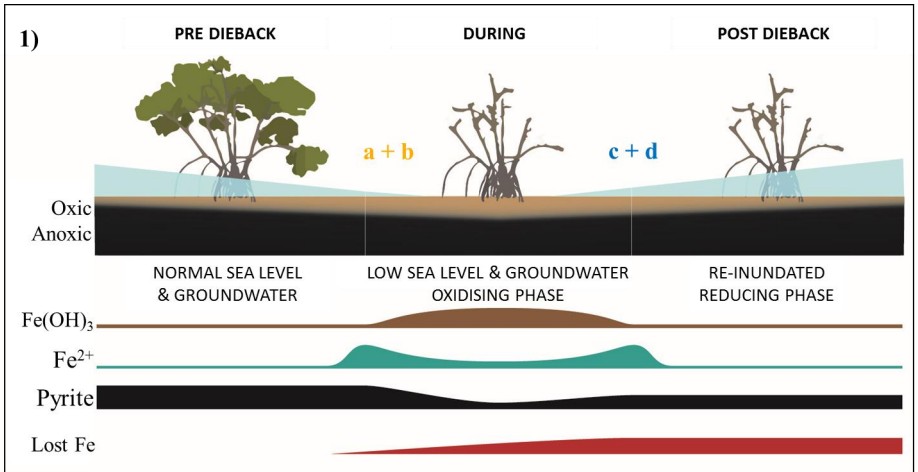

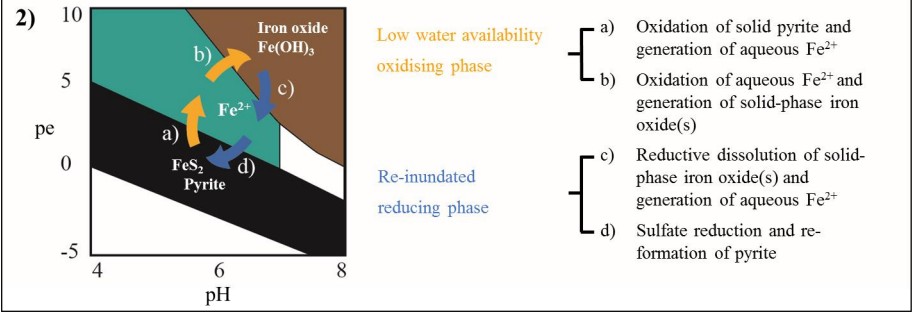


**Figure 9. Conceptual diagram of Fe speciation under different sediment redox, pH conditions and a) how speciation changes would be influenced by sea level and groundwater. Under increased redox conditions b) pyrite oxidation causes Fe transformation to bioavailable $Fe^{2+}$ and particulate $Fe(OH)_3$, consequent reduction of $Fe(OH)_3$**

Our data suggest that the climate-driven changes in sediment geochemistry resulted in
extremely low water availability and drove the mangrove dieback. Mangrove dieback may
also be associated with increased concentrations of bioavailable $Fe^{2+}$ in porewaters that
occurred during this time of low water availability. Estimated losses of Fe from sediments
were consistent with the observed plant uptake and suggest Fe mobilisation due to sediment
oxidation (and subsequent reduction). This Fe mobilisation may also have led to significant
Fe inputs to the ocean. This study supports climate observations suggesting that the Gulf of
Carpentaria dieback was strongly driven by an extreme ENSO event (Harris et al. 2017).
Climate change is increasing the intensity of ENSO events and climate extremes (Lee and
McPhaden 2010, Cai et al. 2014, Freund et al. 2019) and increasing sea level variability
(Widlansky et al. 2015), which is impacting on mangrove forests in arid coastlines (Lovelock
et al. 2017). This study therefore revealed a geochemical mechanism that may also contribute
to mangrove stress and dieback, building on the premise that the dieback event was
associated with climate change  (Harris et al. 2018). Further research is necessary to confirm
the role of Fe in the mortality event, to constrain potential Fe losses to the ocean from
sediments and to understand thresholds for Fe toxicities in *Avicennia marina*.





**Acknowledgements**

JZ Sippo acknowledges funding support and access to ANSTO facilities from AINSE which made this project possible. We would like to thank Jocelyn Turnbull for giving us permission to use atmospheric $^{14}$C data extended to 2017 from Baring Head (Wellington). The study was funded by the Australian Research Council (DE150100581, DP180101285, DE160100443, DP150103286 and LE140100083).





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





**Appendix 1. Cross correlation function (CCF) analysis of the relationship between wood density and**
**climate data over time at one month resolution over a 12 month period prior and post dieback. Wood**
**samples are from the upper, mid and lower intertidal zones of the dead (red) and living (green) mangrove**
**areas. Blue horizontal dashed lines indicate P< 0.01 with n=125. Grey dashed vertical lines at zero lag**
**indicate dieback period.**

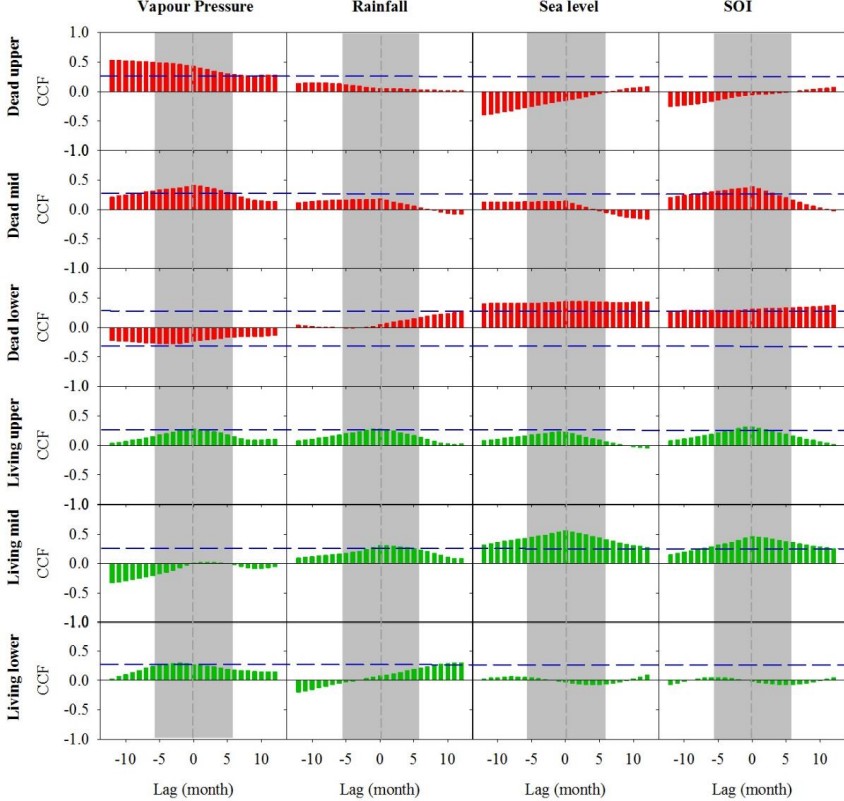
