# Peer review of "Linking climatic-driven iron toxicity and water stress to a massive mangrove dieback"

_Biogeosciences, 2019_

## Referee Comment (RC1) · Anonymous Referee #1 · 10 Feb 2020

Sippo et al. in this manuscript have tried to understand the reasons for massive mangrove dieback in 2015-16 along 1000 km coastline of the Gulf of Carpentaria, Australia. They have analyzed Fe concentrations and water use efficiency from a living and dead mangrove from the region and tried to link it to climate events like ENSO and other parameters such as rainfall, water vapour etc. The manuscript contains a good data set with 14C ages. However, in the end, content of the manuscript does not justify the title. From the beginning, authors have made up their mind that since the Fe content in the dead mangrove is higher than the living, it must be the reason for toxicity and hence the eventual death. From the data, it is quite clear that Fe content is higher in dead mangrove compared to living but at the same time, authors have admitted that there is no report of Fe toxicity at the reported concentration level in this particular species

of mangrove. They have not discussed the physiological aspect of the Fe assimilation by the mangrove. Also, the linkages to the mangrove mortality with climate parameters such as rainfall, sea-level, ENSO etc. comes as a forced attempt. The very fact that these two regions are adjacent to each other with no geomorphic differences (i.e, similar elevation etc.), climatic factors are likely to affect them in almost equal measures. I am not sure if it makes sense to link death of mangroves in one part of the same region to a climatic phenomenon, particularly when it is not affecting the adjacently located mangroves with similar species. Having said that, it remains a fact that mangroves have died in one part and not in the another. I would expect the authors to explore more localised reasons for this dieback. In the end, after discussing regional climate at length, authors themselves have invoked the possible role of groundwater. How the creation of aerobic and anaerobic environments in these two adjacently located patches have varied with time leading to availability of bio-available Fe and higher assimilation of Fe by mangrove remains to be looked into. Moreover, Authors have not provided the information of about the history of tidal regime in the region. Was it different between the living and dead mangroves? From the manuscript it appears that sea level receded from the region leading to oxidation of pyrite and formation of bioavailable Fe leading to assimilation. If this was the case, why only in dieback patch? Also, please keep yourself open for explanation other than Fe toxicity. I think, in general, Fe toxicity is linked to water logging and its likelihood is higher under the anaerobic conditions. Since mangroves are experiencing frequent tidal flooding, they are often anoxic and thus chance of Fe toxicity is normally high. Aeration through specialised roots and other biological activities makes rhizosphere of mangrove species often oxygenated. So, most iron is in oxidized form ($Fe^{3+}$), which is insoluble, forming iron plaque in roots of many mangrove species. Thus, roots of mangroves potentially have high concentration of iron than the stem and leaves. If the tidal flooding is disturbed, oxic zones in mangrove region may increase, which leads the more oxidization condition. Though it favours the oxidation of pyrite and liberate $Fe^{2+}$, most of the $Fe^{2+}$ may quickly oxide to Iron oxyhydroxide due to high aeration. So, during dieback time also, despite the oxidation

of stored pyrite and subsequent increase in sediment iron concentration, availability of bioavailable Fe2+ should be less. Though Iron plaque formation prevent mobilization of toxic metals, due it is high cation affinities it can also block the mobilization of other nutrients. Considering this, during low inundation periods, formation of iron plaque could increase many folds, which in turn affect complete mobilization of other nutrients and ultimately to gradual mortality. In light of above, I would suggest that authors revisit their arguments through physiological aspects of Fe interactions with mangrove and more localized reasons for generation of different situations in adjacently located mangroves. Apart from above, I have following comments: • Abstract needs to be re-written with focus on above comments. The last part pertaining to inputs to ocean and increased productivity appears to overstatement, given that you do not have data to prove so.

Material and method : • This section needs a bit more detail. There are sentences which are repetition. • No information about standards used. • The d13C was directly done on Wood cellulose or it was performed on graphite as in 14C? • CRS was used for what? How is it relevant? • Data analysis contains some sentences already covered in material and methods. • You have used relative concentrations for Fe but later in discuss you invoke absolute concentration level to suggest that present concentration is not enough for dieback? Do not you think that the mention of absolute concentrations would provide a good idea to reader to compare their results if they work on this problem in their region? • The concept of time lag and why was it used needs to be justified. Results : • As mentioned before, it would be a good idea to provide absolute concentration of Fe in wood and sediment. • In Figure 3: why there are less number of data points in living forest of upper and mid intertidal? • Figure 4. No explanation of figure as to how it helps in understanding the discussion. • Fig 6: Here you jump to absolute concentration instead of relative. Also, it would help if you explain the how is it relevant to discussion, probably related to pyrite oxidation. This fact is not coming out clear. Discussion: • Most of the first paragraph appears to be overstatement. For example, it is clear that there is greater assimilation of Fe. How do

you know that it went through the process of pyrite oxidation, particularly when you do not have any data or mechanism to show from this study. • Fe in sediment section: I am inclined to suggest that Fe input to ocean part should be deleted as this is not the primary focus of the manuscript. • Overall, my comments about the discussion remains as above, i.e, to focus on Fe cycling in sediments and look for a relatively localised reason for the mortality. Limitation : I am worried about exactly what authors have put it as a limitation of the study. You cannot claim Fe toxicity as a reason for mangrove mortality and be apprehensive about the whole finding as well.

---

## Short Comment (SC1) · 14 Feb 2020

This manuscript provides an interesting theory: increased Fe toxicity and water stress led to large-scale dieback of mangroves in northern Australia. The authors provide lines of evidence from wood and sediment cores to show that low mean sea level, low water vapour, and low precipitation contributed to changes in the biogeochemistry of the soil, which led to changes in the physiology of the trees. Their data shows a very high increase in wood Fe over the period when the dieback occurred, along with a possible decrease in water use efficiency, and decrease in wood density. However, there were no clear differences between the "dead" site compared to the "live" site, with differences mostly driven by the position in the intertidal (low vs high). Both sites had similar growth rates, similar CRS, WUE, wood density and salinity. The only significant

difference was that the live site had higher Fe+ in the soil that the dead one, a result which appears to be contradictory to their hypothesis. At the moment, the manuscript is written in a way that implies that all the data support their theory, but I am still not convinced. I agree that the climatic conditions led to drastic biogeochemical changes in the soil and mangrove trees, however, this does not explain why some of them died and some of them not. A cause-effect link cannot be established yet.

Overall, I think the data is of high quality and there is potential for it to form the basis of an interesting and novel hypothesis on the effects of drought and sea-level rise on mangrove forests. However, it has to acknowledge that this theory does not prove why mangroves died. The authors mention in the conclusion that differences in groundwater could be the cause of death in some forests, however, they also mention that salinity was similar in both sites. There are still many unanswered questions and the paper needs to be rewritten in a way that provides some answers but also acknowledges that new questions have emerged that are yet to be resolved.

---

## Referee Comment (RC2) · Anonymous Referee #2 · 20 Feb 2020

Although the scientific aspect of the paper by Sippo et al is quite novel, the reasoning of mangrove dieback due to Fe toxicity, drought period and strong ENSO is quite speculative that requires more careful handling before being conclusive.

Since 1985, drought condition during the occurrence of 7 El Nino events affected dead mangroves (34 $\pm$ 1 years old), in contrast to the living mangroves which are younger (21 $\pm$ 4 years old) are survived to 4 El Nino events in Gulf of Carpentaria. Ir was seen that Fe peak in the dead mangrove area at the time of tree mortality were 30 to 90 fold higher than baseline in contrast to the living mangrove which showed an Fe peak 25 fold higher than baseline. Authors argued that low sea level and low rainfall/groundwater reduced soil water content, leading to oxidation of Fe sulphide minerals and release of Fe2+. Fe was observed in the upper intertidal zone. Even in absence

of El Nino event, these Fe bearing-phases dissolution can occur in suboxic conditions in mangrove ecosystems. On the one hand, crab burrow and root system may induce these conditions allowing the renewal of electron acceptors with tides; and on the other hand, physiological activities of mangrove root system can lead to increased O2 concentrations in the sediment (Aquat. Bot. 89 (2), 210–219, 2008). Generally, roots absorb Fe+2 and is highly affected by several plant and environmental factors and their toxicity is often associated with salinity and a low phosphorus or base status of soils. Furthermore, injured leaves or necrotic spots on leaves indicate an accumulation of Fe above 1000 ppm (3 to 6 times as high as the Fe content of healthy leaves). However, the most pronounced symptom is the ratio of Fe to other elements and to heavy metals in particular. The proper Fe:Mn ratio seems to be the most obligatory factor in the tolerance of plants to Fe toxicity.The world average Fe ( $\mu$g g-1) conc in Avicennia marina is estimated to be 54000 (Lewis et al. 2011, Bayen 2012), and 120000 in New Caledonia, the South-West Pacific Ocean (Marchand et al. 2016). The increased uptake of Fe+2 in toxic level by the mangrove root system may reduce growth, DNA damage as evident by morphological or structural damage. Authors could highlight above aspects in their paper by comparing a possible impact of iron and substrate factors on mangrove that would be very relevant in this specific context.

---

## Author Comment (AC1) · 19 May 2020

Comment: From the beginning, authors have made up their mind that since the Fe content in the dead mangrove is higher than the living, it must be the reason for toxicity and hence the eventual death. From the data, it is quite clear that Fe content is higher in dead mangrove compared to living but at the same time, authors have admitted that there is no report of Fe toxicity at the reported concentration level in this particular species C1 of mangrove. They have not discussed the physiological aspect of the Fe assimilation by the mangrove.

Response: We thank the reviewer for this comment and important distinction. We agree that we do not have enough data to directly assess Fe toxicity. We will make

changes throughout the manuscript to clarify that the evidence is strongly suggesting differences in water availability between sites but not necessarily Fe toxicity per se. We will modify the manuscript to use Fe in wood and sediments as an indicator of water availability and the possibility of Fe toxicity will be presented as one (of multiple) possible synergistic stressors. We will also add to the manuscript discussion about the physiological assimilation of Fe in mangroves. For example, Marchand et al. (2016) found that Fe2+ availability can lead to plant uptake and potential toxicity, however Fe2+ uptake in mangroves has rarely been described.

Comment: Also, the linkages to the mangrove mortality with climate parameters such as rainfall, sea-level, ENSO etc. comes as a forced attempt. The very fact that these two regions are adjacent to each other with no geomorphic differences (i.e, similar elevation etc.), climatic factors are likely to affect them in almost equal measures. I am not sure if it makes sense to link death of mangroves in one part of the same region to a climatic phenomenon, particularly when it is not affecting the adjacently located mangroves with similar species. Having said that, it remains a fact that mangroves have died in one part and not in the another. I would expect the authors to explore more localised reasons for this dieback. In the end, after discussing regional climate at length, authors themselves have invoked the possible role of groundwater. How the creation of aerobic and anaerobic environments in these two adjacently located patches have varied with time leading to availability of bio-available Fe and higher assimilation of Fe by mangrove remains to be looked into. Moreover, Authors have not provided the information of about the history of tidal regime in the region. Was it different between the living and dead mangroves? From the manuscript it appears that sea level receded from the region leading to oxidation of pyrite and formation of bioavailable Fe leading to assimilation. If this was the case, why only in dieback patch?

Response: We thank the reviewer for this constructive feedback. We have not focused on explaining why certain areas survived the dieback while others did not. Previous studies (Duke et al., 2017; Harris et al., 2017) provide very strong evidence that water availability in the Gulf of Carpentaria was extremely low prior to and during the dieback event. Here we build of this work with multiple lines of evidence suggesting that changes in sediment geochemistry were also associated with low water availability. Fe concentrations in wood and sediment do suggest that water availability was lower in dead forest areas than living areas. We eliminate elevation as a potential driver of differences in water availability because tree mortality occurred even in the lower inter-tidal zone of dead mangroves which are at the same elevation as the lower intertidal zone of the living forest area. Since other potential water sources: precipitation and tidal flushing are eliminated as being different between the sites, this likely suggests that differences in water availability were driven by regional groundwater flows which are highly spatially variable, For example Stieglitz (2005) highlights that the interrela-tionships between confined and unconfined aquifers in the coastal zone can result in localised differences in groundwater flows.

The discussion regarding climatic drivers were used to assess the likely regional drivers of the dieback as well as linking climatic variability to the observed Fe geochemistry and uptake in trees. We will make these links more explicit in the revised manuscript by clarifying this aspect of the paper. We will also add to this work evidence from a recent publication by Harada et al., (under review) which provides isotopic data of mangrove leaves in dead and living areas of forest at the same study sites. Less enriched leaf $\delta$13C values in living forest areas suggest increased water availability and are consistent with our evidence from sediment and wood chronologies.

Comment: Also, please keep yourself open for explanation other than Fe toxicity. I think, in general, Fe toxicity is linked to water logging and its likelihood is higher un-der the anaerobic conditions. Since mangroves are experiencing frequent tidal flood-ing, they are often anoxic and thus chance of Fe toxicity is normally high. Aeration through specialised roots and other biological activities makes rhizosphere of man-grove species often oxygenated. So, most iron is in oxidized form ($Fe^{3+}$), which is insoluble, forming iron plaque in roots of many mangrove species. Thus, roots of mangroves potentially have high concentration of iron than the stem and leaves. If the tidal flooding is disturbed, oxic zones in mangrove region may increase, which leads the more oxidization condition. Though it favours the oxidation of pyrite and liberate $Fe2+$, most of the $Fe2+$ may quickly oxide to Iron oxyhydroxide due to high aeration. So, during dieback time also, despite the oxidation C2 of stored pyrite and subsequent increase in sediment iron concentration, availability of bioavailable $Fe2+$ should be less. Though Iron plaque formation prevent mobilization of toxic metals, due it is high cation affinities it can also block the mobilization of other nutrients. Considering this, during low inundation periods, formation of iron plaque could increase many folds, which in turn affect complete mobilization of other nutrients and ultimately to gradual mortality. In light of above, I would suggest that authors revisit their arguments through physiological aspects of Fe interactions with mangrove and more localized reasons for generation of different situations in adjacently located mangroves.

Response: We will indeed reduce speculation about Fe toxicity in mangroves and instead focus the manuscript on using Fe as a proxy of water availability, and the role of climate drivers in sediment Fe geochemistry in mangrove ecosystems. The drivers of Fe availability in mangrove sediments are complex and are discussed in detail within the manuscript. We incorporate the suggested discussion of Fe plaque formation and how this may interfere with root uptake of Fe and other minerals.

Comment: Apart from above, I have following comments: âAËŸ c Abstract needs to be ′ re-written with focus on above comments. The last part pertaining to inputs to ocean and increased productivity appears to overstatement, given that you do not have data to prove so.

Response: We will rewrite the abstract in line with the above comment. The statement regarding oceanic productivity changes associate with Fe release will be removed from the abstract.

References

Duke, N. C., J. M. Kovacs, A. D. Griffiths, L. Preece, D. J. Hill, P. Van Oosterzee, J. Mackenzie, H. S. Morning and D. Burrows (2017). "Large-scale dieback of mangroves in Australia's Gulf of Carpentaria: a severe ecosystem response, coincidental with an unusually extreme weather event." Marine and Freshwater Research 68(10): 1816-1829.

Harris, R. M. B., L. J. Beaumont, T. R. Vance, C. R. Tozer, T. A. Remenyi, S. E. Perkins-Kirkpatrick, P. J. Mitchell, A. B. Nicotra, S. McGregor, N. R. Andrew, M. Letnic, M. R. Kearney, T. Wernberg, L. B. Hutley, L. E. Chambers, M. S. Fletcher, M. R. Keatley, C. A. Woodward, G. Williamson, N. C. Duke and D. M. J. S. Bowman (2018). "Biological responses to the press and pulse of climate trends and extreme events." Nature Climate Change 8(7): 579-587.

Marchand, C., J.-M. Fernandez and B. Moreton (2016). "Trace metal geochemistry in mangrove sediments and their transfer to mangrove plants (New Caledonia)." Science of the Total Environment 562: 216-227.

Stieglitz, T. (2005). "Submarine groundwater discharge into the near-shore zone of the Great Barrier Reef, Australia." Marine Pollution Bulletin 51(1): 51-59.

---

## Author Comment (AC2) · 19 May 2020

We thank Fernanda Adame for the constructive feedback on the manuscript and will modify it to clarify the points raised.

Comment from Adame: This manuscript provides an interesting theory: increased Fe toxicity and water stress led to large-scale dieback of mangroves in northern Australia. The authors provide lines of evidence from wood and sediment cores to show that low mean sea level, low water vapour, and low precipitation contributed to changes in the biogeochemistry of the soil, which led to changes in the physiology of the trees. Their data shows a very high increase in wood Fe over the period when the dieback occurred, along with a possible decrease in water use efficiency, and decrease in wood density.

[Figure]

Interactive
comment

However, there were no clear differences between the "dead" site compared to the "live" site, with differences mostly driven by the position in the intertidal (low vs high). Both sites had similar growth rates, similar CRS, WUE, wood density and salinity. The only significant difference was that the live site had higher Fe+ in the soil that the dead one, a result which appears to be contradictory to their hypothesis. At the moment, the manuscript is written in a way that implies that all the data support their theory, but I am still not convinced. I agree that the climatic conditions led to drastic biogeochemical changes in the soil and mangrove trees, however, this does not explain why some of them died and some of them not. A cause-effect link cannot be established yet.

Response: We will re-focus the manuscript to better highlight the differences between dead and living areas. The most obvious difference between areas is Fe concentrations in wood, with peaks in the dead areas which were 30 to 90 fold higher than baseline in contrast to the living mangrove which showed an Fe peak 3-25 fold higher than baseline. If Fe was mobilised during the dieback event, during a period of increased sediment oxidation as hypothesised, then we would expect to see less Fe in the dead mangrove sediments than the living mangrove sediments. This result is therefore in support of the hypothesis (not contradictory) that the dieback is related to water availability. We will make it clear in the manuscript that our results are not conclusive in establishing a cause and effect relationship, but do provide important clues and insights regarding key processes occurring during the mangrove dieback.

Comment from Adame: Overall, I think the data is of high quality and there is potential for it to form the basis of an interesting and novel hypothesis on the effects of drought and sea-level rise on mangrove forests. However, it has to acknowledge that this theory does not prove why mangroves died. The authors mention in the conclusion that differences in groundwater could be the cause of death in some forests, however, they also mention that salinity was similar in both sites. There are still many unanswered questions and the paper needs to be rewritten in a way that provides some answers but also acknowledges that new questions have emerged that are yet to be resolved.

Response: As discussed above, we intend to re-focus the manuscript to have more circumspect conclusions and turn some of the speculation into a new hypothesis. Salinity concentrations in groundwater were taken eight months after the dieback event and may not represent the salinities that occurred during the dieback period. Speculation about the role of groundwater availability in the dieback will also be clarified in consideration of the salinity data.

―――――――――――――――――――――

---

## Author Comment (AC3) · 19 May 2020

Comment from Reviewer 2: Although the scientific aspect of the paper by Sippo et al is quite novel, the reasoning of mangrove dieback due to Fe toxicity, drought period and strong ENSO is quite speculative that requires more careful handling before being conclusive.

Response: We thank the reviewer for this comment. We will make changes throughout the manuscript to avoid excessive speculation. We will clarify that the evidence is strongly suggesting differences in water availability between sites and not necessarily Fe toxicity. We will modify the manuscript to use Fe in wood and sediments as an indicator of water availability and the possibility of Fe toxicity will be presented as one

possible stressor. For example we will change the title of the paper to: "Evidence of climate extremes during a massive mangrove dieback event from wood and sediment chronologies". We will also remove any overly conclusive text from the abstract and manuscript about the relationships between ENSO and Fe concentrations in wood.

Comment from Reviewer 2: Since 1985, drought condition during the occurrence of 7 El Nino events affected dead mangroves (34 $\pm$ 1 years old), in contrast to the living mangroves which are younger (21 $\pm$ 4 years old) are survived to 4 El Nino events in Gulf of Carpentaria. It was seen that Fe peak in the dead mangrove area at the time of tree mortality were 30 to 90 fold higher than baseline in contrast to the living mangrove which showed an Fe peak 25 fold higher than baseline. Authors argued that low sea level and low rainfall/ groundwater reduced soil water content, leading to oxidation of Fe sulphide minerals and release of Fe2+. Fe was observed in the upper intertidal zone. Even in absence of El Nino event, these Fe bearing-phases dissolution can occur in suboxic conditions in mangrove ecosystems.

Response: We will include discussion of how Fe dissolution can occur in suboxic conditions. Importantly we will modify the tone of the manuscript to be less conclusive about the role of Fe toxicity in the forest mortality and instead discuss Fe trends as a reflection of changes in sediment geochemistry over time. Our results of Fe concentrations in wood over time do suggest that a significant change in sediment redox conditions occurred during the period of forest mortality.

Comment from Reviewer 2: On the one hand, crab burrow and root system may induce these conditions allowing the renewal of electron acceptors with tides; and on the other hand, physiological activities of mangrove root system can lead to increased O2 concentrations in the sediment (Aquat. Bot. 89 (2), 210–219, 2008). Generally, roots absorb Fe+2 and is highly affected by several plant and environmental factors and their toxicity is often associated with salinity and a low phosphorus or base status of soils. Furthermore, injured leaves or necrotic spots on leaves indicate an accumulation of Fe above 1000 ppm (3 to 6 times as high as the Fe content of healthy leaves). However,

the most pronounced symptom is the ratio of Fe to other elements and to heavy metals in particular. The proper Fe:Mn ratio seems to be the most obligatory factor in the tolerance of plants to Fe toxicity. The world average Fe ( _g g-1) conc in Avicennia marina is estimated to be 54000 (Lewis et al. 2011, Bayen 2012), and 120000 in New Caledonia, the South-West Pacific Ocean (Marchand et al. 2016). The increased uptake of Fe+2 in toxic level by the mangrove root system may reduce growth, DNA damage as evident by morphological or structural damage. Authors could highlight above aspects in their paper by comparing a possible impact of iron and substrate factors on mangrove that would be very relevant in this specific context.

Response: In response to this constructive comment, we will reduce overstatement of Fe being the cause of mangrove mortality. We will also clarify that the Fe chronologies in wood and sediment are evidence of geochemical changes in the sediments, which also suggest that changes in water availability occurred during the dieback period. We have looked at sediment Fe:Mn ratios in ITRAX data and found no clear differences between living and dead mangrove areas. We assume that this may be because the sediment cores were taken after the dieback period when sediment geochemistry conditions returned to normal. We also looked at Fe:Mn ratios in the wood ITRAX data, these trends overwhelmingly reflect the Fe concentrations. Because we have re-shifted the focus of the manuscript away from Fe toxicity, further exploration of this data may be beyond the scope of this study

---

## Author Response (AR1)

**Response to Editor and Reviewers**

**Associate Editor Decision: Reconsider after major revisions**

While all reviewers recognized the novelty of your manuscript, they were quite critical of your hypothesis on Fe toxicity. I agree with the reviewers and envisage that the manuscript would require a substantial revision to address reviewers' concerns and commensurately refocus data presentation and interpretation. Therefore, I have to recommend 'reconsider after major revisions' and might need to ask reviewers to reassess whether the revised manuscript has provided a reasonable, data-supported interpretation of the observed biogeochemical changes and potential linkage to the mangrove dieback.

**Response:** We thank the editor for the comments and have made extensive changes to the manuscript. The title has changed to prevent the perception of a strong conclusion. The manuscript now focuses more on differences in water availability using Fe as a proxy of sediment redox conditions. Discussion of the possibility of Fe toxicity has been greatly reduced and re-framed accordingly.

As the first reviewer suggested, please pay more attention to providing analytical details, particularly about Itrax core scanner (instrumental information, analytical accuracy, how you processed "semi-quantitative elemental profiles, etc.). In your response to the second reviewer's comment on Fe/Mn ratio, you said that "Because we have re-shifted the focus of the manuscript away from Fe toxicity, further exploration of this data may be beyond the scope of this study." In my view, Mn data would be still helpful for you to clear up any metal toxicity issues. I also wondered if you could compare Fe and other major metals (out of 34 elements) to single out changes in the concentration of Fe at the dieback site from drought-induced (little or small?) changes in the concentrations of other metals.

**Response:** We have made several changes following this comment. See revised methods from L184-190 "Wood samples and sediment cores were analysed for elemental composition with a micro X-ray fluorescence conducted at ANSTO using an Itrax core scanner (Cox Analytical Systems). The scanner produces a high resolution (0.2 mm) radiographic density pattern and semi-quantitative elemental profiles for each sample. The Itrax measured 34 elements and while trends occurred in some elements (see Supplementary Figures 1 & 2), here we focus on Fe. Itrax Fe results have been compared with absolute $Fe_2O_3$ concentrations with high accuracy ($R^2$ = 0.74) (Hunt et al., 2015)".

And also from L202 - 209: "To align radiocarbon calendar ages with Itrax data, we interpolated ages using the wood circumference. Itrax elemental and density data were normalized as the mean subtracted from each value divided by the standard deviation following Hevia et al. (2018) and are referred to hereforth as relative concentrations. We also normalized the Fe data to total counts and other measured elements following Turner et al. (2015) and Gregory et al. (2019) to confirm the trends did not change with different normalization approaches which they did not. This normalization reduces external effects (Gregory et al. 2019) and allows a more direct comparison between samples from living and dead forest areas".

We have particularly examined Mn and Fe/Mn ratio based on your comments and we have added Supplementary Figures 1 & 2 to the manuscript, as well as the following text L420 – 436: "The apparent mobilisation of Fe (loss from sediment and uptake in wood) was not observed in other elements (Supplementary Figures 1 & 2). Sediment Fe:Mn ratios in Itrax data displayed no clear differences between living and dead mangrove areas. These similarities may be because the sediment cores were taken after the dieback period when sediment geochemistry conditions returned to normal. Trends in Mn in the wood samples (Supplementary Figure 1) also show no clear differences between living and dead forest areas and the Fe:Mn ratios in the wood Itrax data overwhelmingly reflect the Fe concentrations".

**Reviewer 1**

**Comment:**

From the beginning, authors have made up their mind that since the Fe content in the dead mangrove is higher than the living, it must be the reason for toxicity and hence the eventual death. From the data, it is quite clear that Fe content is higher in dead mangrove compared to living but at the same time, authors have admitted that there is no report of Fe toxicity at the reported concentration level in this particular species C1 of mangrove. They have not discussed the physiological aspect of the Fe assimilation by the mangrove.

**Response**

We agree that we do not have enough data to directly assess Fe toxicity. We have now made changes throughout the manuscript to clarify that the evidence is strongly suggesting differences in water availability between sites but not necessarily a direct effect of Fe toxicity. For example we have changed the title of the paper to: "Reconstructing extreme climatic and geochemical conditions during the largest natural mangrove dieback on record". We have modified the manuscript to use Fe in wood and sediments as a proxy indicator of water availability and the possibility of Fe toxicity is now presented as one (of multiple) possible synergistic stressors (See marked up manuscript with changes throughout). We have also provided additional background to the manuscript about the physiological assimilation of Fe in mangroves:

L91 – 98: "Marchand et al. (2016) suggest that the presence of Fe2 + may result in an increased Fe uptake by the root system. Such uptake may be toxic for the plant by reducing photosynthesis, increasing oxidative stress, and damaging membranes, DNA and proteins (Marchand et al., 2016). Fe toxicity in some mangrove species is reported to occur at concentrations ~2 fold higher than the optimal Fe supply for maximal growth (Alongi, 2010). However, to our knowledge, Fe toxicity in A. marina at extremely high Fe concentrations has not been investigated."

**Comment**

Section 2 - Also, the linkages to the mangrove mortality with climate parameters such as rainfall, sea-level, ENSO etc. comes as a forced attempt. The very fact that these two regions are adjacent to each other with no geomorphic differences (i.e, similar elevation etc.), climatic factors are likely to affect them in almost equal measures. I am not sure if it makes sense to link death of mangroves in one part of the same region to a climatic phenomenon, particularly when it is not affecting the adjacently located mangroves with similar species. Having said that, it remains a fact that mangroves have died in one part and not in the other. I would expect the authors to explore more localised reasons for this dieback. In the end, after discussing regional climate at length, authors themselves have invoked the possible role of groundwater. How the creation of aerobic and anaerobic environments in these two adjacently located patches have varied with time leading to availability of bio-available Fe and higher assimilation of Fe by mangrove remains to be looked into. Moreover, Authors have not provided the information of about the history of tidal regime in the region. Was it different between the living and dead mangroves? From the manuscript it appears that sea level receded from the region leading to oxidation of pyrite and formation of bioavailable Fe leading to assimilation. If this was the case, why only in dieback patch?

**Response**

Our assessment of the climatic variables and time series analysis was not intended to determine between site differences, but to assess regional scale climate drivers in the area, and whether they provide insight into the mangrove dieback. We have clarified this in the methods L216-217 "Cross correlations with a time lag of one-month intervals were used to evaluate the relationships between regional climatic variables (the Southern Oscillation Index (SOI), sea level, rainfall and vapour pressure) with wood density, elemental relative concentrations and growth rates."

We have now added a section to the discussion to discuss differences in water availability between living and dead forest areas from L519 – 538:

"We have no data to determine if regional groundwater availability was greater in living forest areas than dead forest areas during the mangrove dieback. No significant difference was observed in groundwater salinities 8 months post dieback. However, under normal sea level conditions (i.e. when groundwater samples were collected), tidal inundation is likely to be the predominant driver of groundwater salinities rather than groundwater flows. Duke et al. (2017) and Harris et al. (2017) provide strong evidence that water availability in the Gulf of Carpentaria was extremely low prior to and during the dieback event. In this study we have been able to build on this work by exploring links between changes in sediment geochemistry and low water availability.

We eliminate elevation as a potential driver of water availability in living and dead forest areas. Tree mortality occurred even in the lower intertidal zone of the dead mangrove area which are at the same elevation as the lower intertidal zone of the living forest area (see elevation DEM in Figure 1c). Since other potential water sources were comparable between the sites, differences in water availability were likely driven by groundwater availability. Groundwater flows have high spatial variability and have been demonstrated to be an important water source in mangroves from arid Australia. For example Stieglitz (2005)

highlights that the interrelationships between confined and unconfined aquifers in the coastal zone can result in localised differences in groundwater flows. High resolution spatial analysis of groundwater salinities in living and dead forest areas during low sea level conditions would help to clarify how water sources may drive mangrove mortality".

We have also added evidence from a recent publication providing isotopic data of mangrove leaves in dead and living areas at the same study sites from L364: "Trends in wood density, mangrove growth rate and water use efficiency also reveal distinct differences in water availability between dead and living forest areas. Lower water availability in the dead mangrove forest area was also evident in lower plant growth rates and higher plant water use efficiency. Mangrove plant isotope data at the same sites from a study by Harada et al. (2020) also shows a similar trend with more enriched δ13C values in the dead mangrove zone".

**Comment**

Section 3 - Also, please keep yourself open for explanation other than Fe toxicity. I think, in general, Fe toxicity is linked to water logging and its likelihood is higher under the anaerobic conditions. Since mangroves are experiencing frequent tidal flooding, they are often anoxic and thus chance of Fe toxicity is normally high. Aeration through specialised roots and other biological activities makes rhizosphere of mangrove species often oxygenated. So, most iron is in oxidized form (Fe3+), which is insoluble, forming iron plaque in roots of many mangrove species. Thus, roots of mangroves potentially have high concentration of iron than the stem and leaves. If the tidal flooding is disturbed, oxic zones in mangrove region may increase, which leads the more oxidization condition. Though it favours the oxidation of pyrite and liberate Fe2+, most of the Fe2+ may quickly oxide to Iron oxyhydroxide due to high aeration. So, during dieback time also, despite the oxidation C2 of stored pyrite and subsequent increase in sediment iron concentration, availability of bioavailable Fe2+ should be less. Though Iron plaque formation prevent mobilization of toxic metals, due it is high cation affinities it can also block the mobilization of other nutrients. Considering this, during low inundation periods, formation of iron plaque could increase many folds, which in turn affect complete mobilization of other nutrients and ultimately to gradual mortality. In light of above, I would suggest that authors revisit their arguments through physiological aspects of Fe interactions with mangrove and more localized reasons for generation of different situations in adjacently located mangroves.

**Response**

We agree with the reviewer's general thoughts. However, our observations have not been designed to assess these mechanisms and we feel we should omit excessive speculation following advice from the Associate Editor and other reviewers. We have now reduced speculation about Fe toxicity in mangroves and instead focused the manuscript on using Fe as a proxy of water availability, and the role of climate drivers in sediment Fe geochemistry in mangrove ecosystems. The drivers of Fe availability in mangrove sediments are complex and now discussed within the manuscript from L414 – 422: "While our observations suggest complex sedimentary redox conditions occurred in dead zone mangrove sediments during the dieback event, linking drought and low sea level to porewater Fe concentrations requires further investigation. For example, crab burrows and root systems can induce conditions that increase $O_2$ diffusion into sediments and thus influence $Fe^{2+}$ mobility over tidal cycles (Nielsen at al., 2003; Kirstensen et al., 2008). Localised Fe(III) oxide dissolution can also occur in redox / pH micro-niches and under suboxic conditions (Fabricius et al. 2014; Zhu et al. 2012). Further research on the mechanisms of bioavailable Fe release and the thresholds for Fe toxicities in *Avicennia marina* is required to clearly understand the impacts of porewater Fe on mangrove forests".

**Comment**

Section 4 - Apart from above, I have following comments: âAˇ c Abstract needs to be ´ re-written with focus on above comments. The last part pertaining to inputs to ocean and increased productivity appears to overstatement, given that you do not have data to prove so.

**Response**

We have rewritten the abstract in line with the above comments. The statement regarding oceanic productivity changes associate with Fe release has been removed from the abstract.
* * *
**Specific comments from Reviewer 1**

**Material and method:**
There are sentences which are repetition.
**Response:** Repetitive sentences were removed from the data analysis section.

No information about standards used.
**Response:** Information on standards used for AMS $^{14}C$ analysis was added to the text from L176: "Oxalic acid I (HOxI) was used as the primary standard for calculating sample $^{14}C$ content, while oxalic acid II (HOxII) and IAEA-C7 reference material were used as check standards" .

The d13C was directly done on Wood cellulose or it was performed on graphite as in 14C?
**Response:** This is specified in the manuscript L171: "A portion of graphite was used for the determination of $\delta^{13}C$ for isotopic fractionation correction using a Micromass IsoPrime elemental analyser/isotope ratio mass spectrometer (EA/IRMS) at the Australian Nuclear Science and Technology Organisation (ANSTO)".

CRS was used for what? How is it relevant?
**Response:** We have modified the methods L194-197 to read: "Chromium reducible sulfur (CRS) was measured from sediment samples collected with a Russian peat auger to 1 m depth to provide an estimate of reducible inorganic S (RIS) species such as pyrite ($FeS_2$ - a key oxygen-sensitive sedimentary Fe species) with a linear relationship of $R^2$ = 0.996 (Burton et al., 2008)". We have also added the following text to the discussion to clarify the relevance of CRS from L428: Similar trends were also observed in CRS (a proxy for pyrite) sediment core profiles, which have ~40 % lower $FeS_2$ concentrations in the dead mangroves in comparison to the living mangrove sediments (Figure 6).

Data analysis contains some sentences already covered in material and methods.
**Response:** Repetitive sentences were removed from the data analysis section.

You have used relative concentrations for Fe but later in discuss you invoke absolute concentration level to suggest that present concentration is not enough for dieback? Do not you think that the mention of absolute concentrations would provide a good idea to reader to compare their results if they work on this problem in their region?
**Response:** We have clarified the use of relative and absolute concentration in the data analysis section from L203: "Itrax elemental and density data were normalized as the mean subtracted from each value divided by the standard deviation following the calculation of Z-scores by Hevia et al. (2018) and are referred to hereforth as relative concentrations. Methods that provided absolute concentrations such as CRS are simply referred to as concentrations".

The concept of time lag and why was it used needs to be justified.
**Response:** We have explained the use of time lag analysis in the Data analysis section from L222 – 233: "This time lag analysis was specifically chosen to examine relationships between climate variables and Fe over a two year period because records of all climate variables are in resolution of months, but the chronology of Fe (based on [14]C dates) is in years".

**Results**
As mentioned before, it would be a good idea to provide absolute concentration of Fe in wood and sediment.
**Response:** This comment is addressed above where we have modified the manuscript to clarify the use of both relative and absolute concentrations.

In Figure 3: why there are less number of data points in living forest of upper and mid intertidal?
**Response:** We have explained this occurrence in the manuscript from L247: "Itrax trends are plotted against [14]C ages and since tree growth rates change over time, Itrax data is not evenly distributed over time".

Figure 4. No explanation of figure as to how it helps in understanding the discussion.
**Response:** We feel this is an important Figure which visually shows the relationships between environmental variables and sedimentary Fe relative concentrations over time. We have respectfully left the Figure in the revised version, but we have added text to more fully explain the figure – see line 392-394: "Records of all climate variables are in resolution of months, but the chronology of Fe (based on [14]C dates) is in years. We therefore used time lag analysis to examine relationships between climate variables and Fe over a two year period (Figure 4)".

Fig 6: Here you jump to absolute concentration instead of relative. Also, it would help if you explain the how is it relevant to discussion, probably related to pyrite oxidation. This fact is not coming out clear.

**Response:** The description of Figure 6 in results now reads L298: "Chromium Reduced Sulfur (CRS) absolute concentrations, which provide a proxy for pyrite concentrations in sediment cores, were also lower overall in the dead mangrove compared to the living mangrove area - by 36% in the upper and 38% in the lower intertidal zones respectively (Figure 6)".
* * *
**Reviewer 2**

Although the scientific aspect of the paper by Sippo et al is quite novel, the reasoning of mangrove dieback due to Fe toxicity, drought period and strong ENSO is quite speculative that requires more careful handling before being conclusive.

**Response**

We have made changes throughout the manuscript to avoid excessive speculation. We have clarified that the evidence is strongly suggesting differences in water availability between sites. We have modified the manuscript throughout to use Fe in wood and sediments as an indicator of water availability and the possibility of Fe toxicity has now been presented as one possible stressor. For example we have changed the title of the paper to: "Reconstructing extreme climatic and geochemical conditions during the largest natural mangrove dieback on record". We have also removed any overly conclusive text from the abstract and manuscript about the relationships between ENSO and Fe concentrations in wood.

Since 1985, drought condition during the occurrence of 7 El Nino events affected dead mangroves (34 ± 1 years old), in contrast to the living mangroves which are younger (21 ± 4 years old) are survived to 4 El Nino events in Gulf of Carpentaria. It was seen that Fe peak in the dead mangrove area at the time of tree mortality were 30 to 90 fold higher than baseline in contrast to the living mangrove which showed an Fe peak 25 fold higher than baseline. Authors argued that low sea level and low rainfall/ groundwater reduced soil water content, leading to oxidation of Fe sulphide minerals and release of Fe2+. Fe was observed in the upper intertidal zone. Even in absence of El Nino event, these Fe bearing-phases dissolution can occur in suboxic conditions in mangrove ecosystems.

**Response**

We have modified the tone of the manuscript to be less conclusive about the role of Fe toxicity in the forest mortality and instead discussed Fe trends as a reflection of changes in sediment geochemistry over time. We have now included discussion of how Fe dissolution can occur in suboxic conditions and discussed the complexity of Fe speciation in greater depth from L415 - 423: "While our observations suggest complex sedimentary redox conditions occurred in dead zone mangrove sediments during the dieback event, linking drought and low sea level to porewater Fe concentrations requires further investigation. For example, crab burrows and root systems can induce conditions that increase $O_2$ diffusion into sediments and thus influence $Fe^{2+}$ mobility over tidal cycles (Nielsen at al., 2003; Kirstensen et al., 2008). Localised Fe(III) oxide dissolution can also occur in redox / pH micro-niches and under suboxic conditions (Fabricius et al. 2014; Zhu et al. 2012). Further research on the mechanisms of bioavailable Fe release and the thresholds for Fe toxicities in *Avicennia marina* is required to clearly understand the impacts of porewater Fe on mangrove forests."

On the one hand, crab burrow and root system may induce these conditions allowing the renewal of electron acceptors with tides; and on the other hand, physiological activities of mangrove root system can lead to increased O2 concentrations in the sediment (Aquat. Bot. 89 (2), 210–219, 2008). Generally, roots absorb Fe+2 and is highly affected by several plant and environmental factors and their toxicity is often associated with salinity and a low phosphorus or base status of soils. Furthermore, injured leaves or necrotic spots on leaves indicate an accumulation of Fe above 1000 ppm (3 to 6 times as high as the Fe content of healthy leaves). However, the most pronounced symptom is the ratio of Fe to other elements and to heavy metals in particular. The proper Fe:Mn ratio seems to be the most obligatory factor in the tolerance of plants to Fe toxicity. The world average Fe (g g-1) conc in Avicennia marina is estimated to be 54000 (Lewis et al. 2011, Bayen 2012), and 120000 in New Caledonia, the South-West Pacific Ocean (Marchand et al. 2016). The increased uptake of Fe+2 in toxic level by the mangrove root system may reduce growth, DNA damage as evident by morphological or structural damage. Authors could highlight above aspects in their paper by comparing a possible impact of iron and substrate factors on mangrove that would be very relevant in this specific context.

**Response**

We have removed the perceived overstatement that Fe was the cause of mangrove mortality throughout the manuscript. We have also clarified that the Fe chronologies in wood and sediment are evidence of geochemical changes in the sediments, which also suggest that changes in water availability occurred during the dieback period. We have now provided Mn data from wood samples in Supplementary Figure 1 and from Ca in Supplementary Figure 2 and we have added the following text to the manuscript L438 – 444: "The apparent mobilisation of Fe (loss from sediment and uptake in wood) was not observed in other elements (Supplementary Figures 1 & 2). Sediment Fe:Mn ratios in Itrax data displayed no clear differences between living and dead mangrove areas. These similarities may be because the sediment cores were taken after the dieback period when sediment geochemistry conditions returned to normal. Trends in Mn in the wood samples (Supplementary Figure 1) also show no clear differences between living and dead forest areas and the Fe:Mn ratios in the wood Itrax data overwhelmingly reflect the Fe concentrations.".
* * *
**Reviewer 3 - Fernanda Adame**

We thank Dr. Fernanda Adame for the constructive feedback on the manuscript and have addressed the comments she raised.

This manuscript provides an interesting theory: increased Fe toxicity and water stress led to large-scale dieback of mangroves in northern Australia. The authors provide lines of evidence from wood and sediment cores to show that low mean sea level, low water vapour, and low precipitation contributed to changes in the biogeochemistry of the soil, which led to changes in the physiology of the trees. Their data shows a very high increase in wood Fe over the period when the dieback occurred, along with a possible decrease in water use efficiency, and decrease in wood density. However, there were no clear differences between the "dead" site compared to the "live" site, with differences mostly driven by the position in the intertidal (low vs high). Both sites had similar growth rates, similar CRS, WUE, wood density and salinity. The only significant difference was that the live site had higher Fe+ in the soil that the dead one, a result which appears to be contradictory to their hypothesis. At the moment, the manuscript is written in a way that implies that all the data support their theory, but I am still not convinced. I agree that the climatic conditions led to drastic biogeochemical changes in the soil and mangrove trees, however, this does not explain why some of them died and some of them not. A cause-effect link cannot be established yet.

**Response**

As noted in earlier responses, we have re-focused the manuscript to better highlight the differences between dead and living areas with a new section in the discussion titled 'Differences in water availability between living and dead forest areas' from L490 - 519. We have also made it clear throughout the manuscript that our results are not conclusive in establishing a cause and effect relationship, but do provide important clues and insights regarding key processes occurring during the mangrove dieback.

Overall, I think the data is of high quality and there is potential for it to form the basis of an interesting and novel hypothesis on the effects of drought and sea-level rise on mangrove forests. However, it has to acknowledge that this theory does not prove why mangroves died. The authors mention in the conclusion that differences in groundwater could be the cause of death in some forests, however, they also mention that salinity was similar in both sites. There are still many unanswered questions and the paper needs to be rewritten in a way that provides some answers but also acknowledges that new questions have emerged that are yet to be resolved.

**Response**

As discussed above, we have now re-focused the manuscript to have more circumspect conclusions and turn some of the speculation into a new hypothesis. Salinity concentrations in groundwater were taken eight months after the dieback event and may not represent the salinities that occurred during the dieback period. We have clarified this in the manuscript L520-524: "We have no data to determine if regional groundwater availability was greater in living forest areas than dead forest areas during the mangrove dieback. No significant difference was observed in groundwater salinities 8 months post dieback. However, under normal sea level conditions (i.e. when groundwater samples were collected), tidal inundation is likely to be the predominant driver of groundwater salinities rather than groundwater flows".

END OF RESPONSES

Reconstructing extreme climatic  and
geochemical conditions during the largest natural mangrove dieback on record

James Z. Sippo[1,2], Isaac R. Santos[2,3], Christian J. Sanders[2], Patricia Gadd[4], Quan Hua[4], Catherine E. Lovelock[5],
Nadia S. Santini[6,7], Scott G. Johnston[1], Yota Harada[8], Gloria Reithmeir[1], Damien T. Maher[1,9]

[1]Southern Cross Geoscience, Southern Cross University, Lismore, 2480 Australia.
[2]National Marine Science Centre, Southern Cross University, PO Box 4321, Coffs Harbour,
NSW 2450, Australia
[3]Department of Marine Sciences, University of Gothenburg, Sweden
[4]Australian Nuclear Science and Technology Organisation (ANSTO), Locked Bag 2001,
Kirrawee DC, NSW 2232, Australia
[5]School of Biological Sciences, the University of Queensland, St Lucia QLD 4072, Australia
[6]Cátedra Consejo Nacional de Ciencia y Tecnología, Av. Insurgentes Sur 1582, Crédito
Constructor, Benito Juárez, 03940, Ciudad de México, Mexico.
[7]Instituto de Ecología, Universidad Nacional Autónoma de México, Ciudad Universitaria,
04500, Ciudad de México, Mexico.
[8]Australian Rivers Institute – Coast and Estuaries, and School of Environment and Science,
Griffith University, Gold Coast, QLD 4222, Australia
[9]School of Environment, Science and Engineering, Southern Cross University, Lismore 2480,
Australia

**Abstract**

*A massive mangrove dieback event occurred in 2015/2016 along ~1000km of pristine*
*coastline in the Gulf of Carpentaria, Australia.* Here, we
use *sediment and wood chronologies to*  gain insights into *geochemical and*
*climatic changes.* related to this dieback. *The unique combination of low rainfall and low sea*
*level observed during the dieback event was unprecedented in the previous three decades.*
A combination of *iron (Fe) chronologies in wood and*
*sediment, wood* density *and* estimates of *mangrove water use efficiency* ,
*all imply lower water availability within the dead mangrove forest. Wood and sediment*
*chronologies suggest a rapid* , *large mobilization of sedimentary Fe, which*
is
consistent with redox transitions promoted by changes in soil moisture content. Elemental
*analysis of wood cross sections revealed 30-90 fold increase in Fe concentrations in dead*
mangroves *just prior to* their *mortality.* Mangrove *wood*
uptake of Fe
*during the dieback*
is consistent with large apparent
losses of Fe from sediments, *which* potentially caused an outwelling *of Fe to* the
ocean . Although Fe toxicity may also have played a role in

*the dieback, this possibility requires further study. We suggest that differences in wood and*
*sedimentary Fe between living and dead forest areas reflect sediment redox transitions that*
*are in turn associated with regional variability in groundwater flows.* Overall, our
observations  provide multiple lines of evidence that the forest dieback
was associated with low water availability , coinciding with a strong
ENSO event.

**Introduction**

Mangroves provide a wide range of ecosystem services , including nursery habitat,
carbon sequestration, and coastal protection (Barbier et al. 2011, Donato et al. 2011). Climate
change is a major threat to mangroves , which adds to existing stressors
imposed by deforestation and over-exploitation (Hamilton and Casey 2016, Richards and
Friess 2016). Sea level rise, altered sediment budgets, reduced
water availability and increasing climatic extremes are all negatively affecting mangroves
(Gilman et al. 2008, Alongi 2015, Lovelock et al. 2015, Sippo et al. 2018).

In Australia, an extensive mangrove dieback event  in the Gulf of Carpentaria
during December 2015 - January 2016, coincided with extreme drought and low
regional sea levels.  This extreme climatic event drove the largest recorded mangrove
mortality event (~1000 km coastline, ~7400 ha) attributed to natural causes (Duke
et al., 2017; Harris et al., 2017; Sippo et al., 2018) and led to extensive changes in the
coastal carbon cycle (Sippo et al. 2019; Sippo et al. 2020) and coastal foodwebs (Harada et
al. 2020). Two other large scale mangrove dieback events occurred at the same time
, one in Exmouth (Lovelock et al. 2017) and
the other in Kakadu National Park,  Australia (Asbridge
et al. 2019).

Mangrove mortality has been previously attributed to low water availability
associated with extreme drought. Limited rainfall and groundwater availability
combined with anomalously low sea levels, effectively reduced tidal inundation and
soil water content (Duke et al., 2017; Harris et al., 2017). A strong El Niño event resulted in
the lowest recorded rainfall in the nine months preceding the mangrove dieback since 1971,
and was accompanied by regional sea levels that were 20 cm lower than average
(Harris et al., 2017). Atmospheric moisture was also unusually low during 2015 - a feature
which may influence the physiological functioning of mangrove trees (Nguyen et al. 2017).
Such severe climatic and hydrologic changes may affect both plant physiology and
sediment geochemistry.

In contrast to terrestrial forest soils, mangrove sediments are largely anoxic due to their
water-logged nature, and high organic matter contents. Mangrove sediments also receive a
supply of materials from both terrestrial environments (e.g. Fe, sediments) and oceanic water
(e.g. $SO_4$) which results in distinctly different biogeochemical cycling
than terrestrial forests (Burdige 2011). As a result, mangrove sediments often accumulate
substantial (~1-5%) bioauthigenic pyrite ($FeS_2$). Pyrite remains stable under
waterlogged and reducing conditions (van Breemen 1988, Johnston et al. 2011).
However, lowering of water levels can alter sediment redox conditions and result in
rapid oxidation of $FeS_2$, releasing acid and dissolved Fe (mostly as more soluble $Fe^{2+}$±
species) to porewaters (Burton et al. 2006, Johnston et al. 2011, Keene et al. 2014).
Subsequent oxidation of $Fe^{2+}$ and precipitation of Fe(III) (oxy)hydroxide minerals can
then lead to the accumulation of highly reactive Fe in sediments. Such reactive
Fe(III) minerals are in turn readily subject to reductive dissolution and (re)-formation of
soluble $Fe^{2+}$ species during any subsequent switch to more reducing conditions.
Thus, changes in sediment redox conditions (e.g. increased oxidation and followed by subsequent reduction, in mangrove sediments that are rich in $FeS_2$

can cause a release of relatively mobile and bioavailable $Fe^{2+}$  during the redox transition(s).

Mobilisation of Fe due to fluctuating oxidation/reduction cycles could also have important consequences for coastal Fe cycling. For example, Fe is often a limiting nutrient in ocean surface waters and thus Fe outwelling from mangroves could have important implications for primary productivity in coastal zone waters (Jickells and Spokes , Fung et al. 2000, Holloway et al. 2016). Fe mobilisation also means that uptake of $Fe^{2+}$ into mangrove tissues may be a powerful proxy for historic sediment redox conditions

. However, the process of  Fe assimilation into mangrove tissues remains poorly understood. Marchand et al. (2016)

suggest that the presence of $Fe^{2+}$ may  result in an increased Fe uptake by the root system. Such uptake may be toxic for the plant by reducing photosynthesis, increasing oxidative stress, and damaging membranes, DNA and proteins (Marchand et al., 2016). Fe toxicity in some mangrove species is reported to occur at concentrations ~2 fold higher than the optimal Fe supply for maximal growth (Alongi, 2010).

However, to our knowledge,

Fe toxicity in *Avicenia marina* at extremely high Fe concentrations has not been investigated.

An extensive saltmarsh dieback in southern United States in 2000 provides an analogue to the mangrove dieback studied here. The saltmarsh dieback coincided with severe drought conditions (McKee et al. 2004, Ogburn and Alber 2006, Alber et al. 2008). McKee et al.

(2004) found that sediments in dead saltmarsh areas had significantly higher acidity upon oxidation than alive areas. The dieback may have been caused by a combination of reduced water availability, increased sediment salinities and/or metal toxicity associated with soil acidification following sediment pyrite oxidation. However, the precise cause of the dieback is a matter of debate and remains inconclusive (McKee et al.

2004, Silliman et al. 2005, Alber et al. 2008). In contrast to the herbaceous salt marsh species affected in the US dieback, mangroves are woody - thus providing opportunity for dendrochronological climatic reconstruction (Verheyden et al. 2005,

[revised manuscript text omitted]

Wood samples were dated using bomb $^{14}$C (eg, Santini et al. 2013; Witt et al. 2017). Water-use efficiency (WUE), which is the ratio of net photosynthesis to transpiration, was assessed using wood cellulose stable isotopic composition $\delta^{13}$C following (Van Der Sleen et al., (2015) as water use efficiency correlates with  $\delta^{13}$C (Farquhar and Richards 1984, Farquhar et al. 1989).  Sub-samples for $^{14}$C and $\delta^{13}$C were taken from tree samples (wood disks) along the longest radius of each disk at regular intervals from the centre to the outer edge (youngest wood). The sub-samples were collected using a scalpel parallel to tree rings to reduce errors. Alpha cellulose was extracted from the wooden sub-samples (Hua et al., 2004b), combusted to $CO_2$ and converted to graphite (Hua et al., 2001). A portion of graphite was used for the determination of $\delta^{13}C$ for isotopic fractionation correction using a Micromass IsoPrime elemental analyser/isotope ratio mass spectrometer (EA/IRMS) at the Australian Nuclear Science and Technology Organisation (ANSTO). The remaining graphite was analysed for $^{14}C$ using the STAR accelerator mass spectrometry (AMS) facility at ANSTO (Fink et al. 2004) with a typical analytical precision of better than 0.3% ($2\sigma$). Oxalic acid I (HOxI) was used as the primary standard for calculating sample $^{14}C$ content, while oxalic acid II (HOxII) and IAEA-C7 reference material were used as check standards. Sample $^{14}C$ content was converted to calendar ages using the ''Simple Sequence'' deposition model of the OxCal calibration program based on chronological ordering (outer samples are younger than inner samples) (Bronk Ramsey, 2008), and atmospheric $^{14}C$ data from Baring Head (Wellington, New Zealand) extended to 2017.

Wood samples and sediment cores were analysed for elemental composition with a micro X-ray fluorescence conducted at ANSTO using an Itrax core scanner (Cox Analytical Systems). The scanner produces a high resolution (0.2 mm) radiographic density pattern and semi-quantitative elemental profiles for each sample. The Itrax measured 34 elements, and while trends occurred in some elements (see Appendix 1 & 2), here we focus on Fe. Itrax Fe results have been compared with absolute $Fe_2O_3$ concentrations with high accuracy ($R^2 = 0.74$) (Hunt et al., 2015). Wood samples were scanned along the same transect as for $^{14}C$ samples, i.e. the longest radius from the wood core to the outer edge. Sediment cores were analysed using the Itrax in four 50 cm increments. Immediately upon collection, CRS sub-samples were placed in polyethylene bags with air removed and frozen prior to CRS analysis. Chromium reducible sulfur (CRS) was measured at 5 cm intervals to 1 m depth to provide an estimate of reducible inorganic S (RIS) species such as pyrite ($FeS_2$ - a key oxygen-sensitive sedimentary Fe species) with a linear relationship of $R^2 = 0.996$ (Burton et al., 2008). Groundwater salinity values were taken at the same sites as wood samples from bore holes dug to ~1m depth. Groundwater in the holes was purged and allowed to refill and salinities were measured using a Hach multi-sonde.

*Data analysis*

To align radiocarbon calendar ages with Itrax data, we interpolated ages using the wood circumference. Itrax elemental and density data were normalized as the mean subtracted from each value divided by the standard deviation following Hevia et al. (2018) and are referred to hereforth as relative concentrations. We also normalized the Fe data to total counts and other measured elements following Turner et al. (2015) and Gregory et al. (2019) to confirm the trends did not change with different normalization approaches which they did not. This normalization reduces external effects (Gregory et al. 2019) and allows a more direct comparison between samples from living and dead forest areas. Methods that provided absolute concentrations such as CRS are simply referred to as concentrations. Growth rates in mm per year were calculated as the measured increment divided by the difference in years (estimated from $^{14}C$) between samples. De-trended growth rates were then calculated as the deviation from the exponential curve fitted to growth rates for each sample.
Water use efficiency (WUE) was calculated from $\delta^{13}C$ isotope values (Van Der Sleen et al.
2015). Differences in WUE between living and dead mangrove areas were compared using T-
test.

[revised manuscript text omitted]

Chromium Reduced Sulfur (CRS) absolute concentrations, which provide a proxy for pyrite concentrations in sediment cores, were also lower overall in the dead than mangrove compared to the living mangrove area - by 36% in the upper and 38% in the lower intertidal zones. These respectively (Figure 6). Although these differences were not significant (Mann-Whitney Rank Sum Test, P > 0.05) but), they were very similar to Itrax differences (Figure 6). Dead mangrove sedimentFe trends. In the upper intertidal zone, CRS concentrations generally increased from ~10 cm depth. Living mangrove CRS increased from ~30 cm depth compared to lower levels deeperwith depth, while in the soil profile, implying some recent CRS loss. In the lower intertidal zone, CRS concentrations were highestpeaked from ~10 cm below the surface in both dead and living sediment samples and then decreased with depth.

 Differences in CRS concentrations (in both the upper and lower intertidal zones) between the dead and living mangroves were most prominent in the upper ~60 cm of each core and tended to converge at greater depths (Figure 6).

[Figure]

**Figure 6**. **Chromium reducible sulfur (CRS) profiles (a proxy for pyrite) from sediment cores in dead (red) and living (green) mangrove areas in the Gulf of Carpentaria.**

Water use efficiency (WUE) calculated from $\delta^{13}C$ decreased in all wood samples from 1983 to 2017 (Figure 7a), suggesting increasing water availability in the study area. During the dieback event, median WUE values were higher in dead samples than in living samples, with the differences more pronounced in the upper intertidal zones (Figure 7b). Comparison of WUE in dead and living mangrove samples suggests lower water availability in the dead mangrove area (Figure 7b). However, the mean WUE values were compared from 1983 to 2017 and were not significantly different (T-test, P = 0.2) in dead and living mangrove areas. Groundwater salinity values were highest in the upper intertidal mangrove areas and lowest in the lower intertidal areas (Figure 7c). Salinities were not significantly different in the living and dead forest areas (T-test, P = 0.913).

[Figure]

**Figure 7. a) Changes to Water Use Efficiency (WUE) over time in wood samples collected from the upper,**
**lower and mid intertidal zone in living (green) and dead (red) mangrove areas. The grey bar represents**
**the mangrove dieback event. Error bars  are not visible due to low error of**
**individual samples. b) Box plot of water use efficiency in mangrove wood samples in dead and living**
**mangrove areas in the upper, mid and lower intertidal zones. Sample size > 4 from each wood sample.**
**The central horizontal line represents the median, the box represents the upper and lower quartiles, and**
**the whiskers represent the maximum and minimum values. c) Box plot of groundwater salinity eight**
**months post dieback event in dead and living mangrove areas in the upper, mid and lower intertidal**
**zones. Sample size > 3 from each intertidal zone.**

Normalised wood density values in the dead mangrove forest showed no change during the dieback event in the upper intertidal zone, but a decline in density values occurred in the mid and lower intertidal zones (Figure 8). In the living mangrove area, declines in wood density values occurred in the upper and mid intertidal zones during the mortality event, but no variation in density occurred in the lower intertidal zone (Figure 8).

[Figure]

**Figure 8. Normalised wood density (relative concentrations) in mangrove wood over time in living (green dots) and dead (red dots) mangrove areas of the Gulf of Carpentaria, Australia. The grey bar represents the time period of the dieback event.**

**Discussion**

**Evidence of differences in water availability between living and dead forest areas from dendrogeochemistry**

Multiple lines of evidence from wood samples and sediment cores point to substantial differences in water availability between the dead and living mangrove areas. For example, Fe trends in wood (comparative Fe gain) and sediment samples (comparative Fe loss) (Figures 3, 5 and 6) within the dead mangrove zone, both suggest the mobilisation of bioavailable Fe as $Fe^{2+}$  .These observations are consistent with oscillations in sedimentary redox conditions, triggered by changes in water availability, promoting mobilisation of Fe - firstly as bioauthigenic pyrite is oxidised and yet again during the reduction of Fe(III) oxide species when conditions return to being predominantly anaerobic (Figure 9). Increased oxygen diffusion into sediments during the period of low water availability likely resulted in the oxidation of bioauthigenic pyrite, which transformed into aqueous and bioavailable $Fe^{2+}$ (e.g. Figure 9.2a; Johnston et al. 2011). With further oxidation, $Fe^{2+}$ would likely have transformed into solid-phase Fe(III)oxides (Figure 9.2b). Such Fe(III) oxides are highly reactive and thus any subsequent short-term reduction (e.g. due to tidal inundation) would also result in re-mobilization of Fe as $Fe^{2+}$ (Figure 9.2c). The fact that these trends in Fe that were observed in wood and soil samples were not observed for other elements analyzed by Itrax, supports the hypothesis that Fe trends were likely related to pyrite oxidation / redox oscillations (Appendix 1 & 2).

The most probable cause for a shift from reducing to oxidising conditions in the sediment is a reduction in water content (Keene et al. 2014) associated with the intense El Niño of 2015/16 and associated low sea level and annual rainfall (Figure 2). Trends in wood density, mangrove growth rate and water use efficiency also reveal distinct differences in water availability between dead and living forest areas. Lower water availability in the dead mangrove forest area was also evident in lower plant growth rates and higher plant water use efficiency. Mangrove plant isotope data at the same sites from a study by Harada et al. (2020) also shows a similar trend with more enriched $\delta^{13}C$ values in the dead mangrove zone.

[Figure]

[Figure]

**Figure 9. Conceptual diagram of Fe speciation under different sediment redox, pH conditions and 1) how speciation changes would be influenced by sea level and groundwater. Under initially elevated redox conditions due to low water availability 2) pyrite oxidation causes Fe transformation to (a) bioavailable $Fe^{2+}$ and (b) particulate $Fe(OH)_3$ , followed by eventual re-establishment of normal water availability / reducing conditions and (c) consequent reduction of $Fe(OH)_3$ and generation of $Fe^{2+}$ followed by (d) sequestration of Fe(II) species via pyrite reformation.**

*Fe in wood* -

Elemental composition from wood samples suggest that the mangrove forest experienced sharp changes in sediment geochemistry during the dieback phase (Figure 3). This is consistent with low sea level and low rainfall/groundwater reducing soil water content, leading to oxidation of Fe sulphide minerals and release of $Fe^{2+}$+ (Figure 9.2a). The Fe peaks in the dead mangrove area at the time of tree mortality were 30 to 90 fold higher than baseline Fe (the mean Fe concentration in the sample prior to the dieback event).

In the living mangrove area, an Fe peak 25 fold higher than baseline Fe was observed in the upper intertidal zone (Figure 3). In the mid and lower intertidal areas of the living mangroves, Fe peaks were 4 and 3 fold higher than baseline respectively. In all living wood samples, Fe subsequently decreased after the dieback event, thereby suggesting that Fe in new wood growth was diminished in association with a return to sustained reducing sediment conditions and a concomitant attenuation in porewater Fe$^{2+}$ availability (Figure 9.2 d).

Records of all climate variables are in resolution of months, but the chronology of Fe (based on $^{14}$C dates) is in years. We therefore used time lag analysis to examine relationships between climate variables and Fe over a two year period (Figure 4). Fe wood concentrations were significantly correlated with both rainfall and vapour pressure in the dead and living forest areas (Figure 4). However, because all climate variables were strongly correlated to each other, we cannot separate the relationships between individual climate drivers and Fe trends. We speculate that the combination of low availability of fresh groundwater and low sea level during the strong El Niño event of 2015/16 are key drivers of the sediment redox conditions , as reflected in wood Fe trends.

Considering the extreme increases in Fe concentrations observed in the wood samples during the dieback event, it is plausible that Fe toxicity could have contributed to mangrove mortality. However, we cannot fully test this hypothesis in this study and are unaware of research testing the toxicity of Fe in *A. marina* at highly elevated concentrations of bioavailable Fe$^{2+}$. Alongi, (2010) found that Fe toxicity occurred in some mangrove species at high concentrations (100 mmol m$^2$ d$^{-1}$ of water-soluble

Fe-EDTA) that were approximately 2 fold higher than the Fe supply for maximal growth.

However, *A. marina* (the dominant species affected by the dieback at the study site)

appear relatively resilient to high porewater Fe$^{2+}$. For example, Johnston et al. (2016)

observed no *A. marina* mortality at porewater Fe$^{2+}$ concentrations of 7-15 fold above normal in a mangrove forest impacted by acid sulfate drainage. Considering that other mangrove species are affected by Fe toxicity at twofold the optimal Fe availability, it is possible that a 30-90 fold increase in Fe could have been an additional stressor to mangroves already stressed by low water availability.

While our observations suggest complex sedimentary redox conditions occurred in dead zone mangrove sediments during the dieback event, linking drought and low sea level to porewater Fe concentrations requires further investigation. For example, crab burrows and root systems can induce conditions that increase O$_2$ diffusion into sediments and thus influence Fe$^{2+}$ mobility over tidal cycles (Nielsen at al., 2003; Kirstensen et al., 2008). Localised Fe(III) oxide dissolution can also occur in redox / pH micro-niches and under suboxic conditions (Fabricius et al. 2014; Zhu et al. 2012). Further research on the mechanisms of bioavailable Fe release and the thresholds for Fe toxicities in *A. marina* is required to clearly understand the impacts of porewater Fe on mangrove forests.

***Fe in sediments***

Sediment cores also displayed considerable differences in down core Fe profiles between living and dead mangrove areas (Figure 5a and b). Normalized Fe concentrations were lower in the upper 1 m of sediments in the dead mangrove area compared to the living, but were very similar in sediments deeper than 1 m (Figure 5a and b). Similar trends were also observed in CRS (a proxy for pyrite, FeS$_2$) sediment core profiles, which have ~40 % lower

FeS$_2$ concentrations in the dead mangroves in the upper 60 cm of the profile, in comparison to the living mangrove sediments (Figure 6). The fact that differences in down core trends in Fe are most prominent in the upper parts of the sediment
cores is consistent with decreases
in water availability being more confined to the upper parts of the sediment
profile, whereas deeper sediments are
more likely to have remained
fully saturated.

Although mangrove sediment conditions are typically highly heterogeneous (Zhu et al. 2006,
Zhu and Aller 2012), the sediment core results are broadly consistent with the wood data. The
apparent mobilisation of Fe (loss from sediment and uptake in wood) was not observed in
other elements (Appendix 1 & 2). Sediment Fe:Mn ratios in Itrax data displayed no clear
differences between living and dead mangrove areas. These similarities may be because the
sediment cores were taken after the dieback period when sediment geochemistry conditions
returned to normal. Trends in Mn in the wood samples (Appendix 1) also show no clear
differences between living and dead forest areas and the Fe:Mn ratios in the wood Itrax data
overwhelmingly reflect the Fe concentrations.

[revised manuscript text omitted]

*Differences in water availability between living and dead forest areas*

We have no data to determine if regional groundwater availability was greater in living forest areas than dead forest areas during the mangrove dieback. No significant difference was observed in groundwater salinities 8 months post dieback. However, under normal sea level conditions (i.e. when groundwater samples were collected), tidal inundation is likely to be the predominant driver of groundwater salinities rather than groundwater flows. Duke et al. (2017) and Harris et al. (2017) provide strong evidence that water availability in the Gulf of Carpentaria was extremely low prior to and during the dieback event. In this study we have been able to build on this work by exploring links between changes in sediment geochemistry and low water availability.

We eliminate elevation as a potential driver of water availability in living and dead forest areas. Tree mortality occurred even in the lower intertidal zone of the dead mangrove area which are at the same elevation as the lower intertidal zone of the living forest area (see elevation DEM in Figure 1c). Since other potential water sources were comparable between the sites, differences in water availability were likely driven by groundwater availability. Groundwater flows have high spatial variability and have been demonstrated to be an important water source in mangroves from arid Australia. For example Stieglitz (2005) highlights that the interrelationships between confined and unconfined aquifers in the coastal zone can result in localised differences in groundwater flows. High resolution spatial analysis of groundwater salinities in living and dead forest areas during low sea level conditions would help to clarify how water sources may drive mangrove mortality.

[revised manuscript text omitted]

Fabricius, A.-L., Duester, L., Ecker, D., Ternes, T.A., 2014. "New Microprofiling and Micro Sampling System for Water Saturated Environmental Boundary Layers". *Environmental Science & Technology*, **48**(14): 8053-8061.

Farquhar, G., K. Hubick, A. Condon and R. Richards (1989). Carbon isotope fractionation and plant water-use efficiency. *Stable isotopes in ecological research*, Springer**:** 21-40.

Farquhar, G. and R. Richards (1984). "Isotopic composition of plant carbon correlates with water-use efficiency of wheat genotypes." *Funct. Plant Biol.* **11**(6): 539-552.

Fink, D., M. Hotchkis, Q. Hua, G. Jacobsen, A. M. Smith, U. Zoppi, D. Child, C. Mifsud, H. van der Gaast and A. Williams (2004). "The antares AMS facility at ANSTO." *Nuclear Instruments and Methods in Physics Research Section B: Beam Interactions with Materials and Atoms* **223**: 109-115.

Freund, M. B., B. J. Henley, D. J. Karoly, H. V. McGregor, N. J. Abram and D. Dommenget (2019). "Higher frequency of Central Pacific El Niño events in recent decades relative to past centuries." *Nature Geoscience* **12**(6): 450-455.

Fung, I. Y., S. K. Meyn, I. Tegen, S. C. Doney, J. G. John and J. K. Bishop (2000). "Iron supply and demand in the upper ocean." *Global Biogeochem. Cycles* **14**(1): 281-295.

Google Earth (2019). "Karumba, Qld, Australia". Digital Globe

Gilman, E. L., J. Ellison, N. C. Duke and C. Field (2008). "Threats to mangroves from climate change and adaptation options: A review." *Aquat. Bot.* **89**(2): 237-250.

Gregory, B. R. B., R. T. Patterson, E. G. Reinhardt, J. M. Galloway and H. M. Roe (2019). "An evaluation of methodologies for calibrating Itrax X-ray fluorescence counts with ICP-MS concentration data for discrete sediment samples." *Chemical Geology* **521**: 12-27.

Hamilton, S. E. and D. Casey (2016). "Creation of a high spatio-temporal resolution global
database of continuous mangrove forest cover for the 21st century (CGMFC-21)." *Global*
*Ecol. Biogeogr.* **25**(6): 729-738.

Harada, Y., R. M. Connolly, B. Fry, D. T. Maher, J. Z. Sippo, L. C. Jeffrey, A. J. Bourke and
S. Y. Lee (2020). "Stable isotopes track the ecological and biogeochemical legacy of mass
mangrove forest dieback in the Gulf of Carpentaria, Australia." *Biogeosciences Discuss.*
**2020**: 1-32.

[revised manuscript text omitted]

Sippo, J. Z., D. T. Maher, K. G. Schulz, C. J. Sanders, A. McMahon, J. Tucker and I. R.
Santos (2019). "Carbon outwelling across the shelf following a massive mangrove dieback in
Australia: Insights from radium isotopes." *Geochimica et Cosmochimica Acta* **253**: 142-158.

Sippo, J. Z., C.J. Sanders, I. R. Santos, L.C. Jeffrey, M. Call, Y. Harada, K. Maguire, D.
Brown, S.R. Conrad and D. T. Maher (2019). " Coastal carbon cycle changes following
mangrove loss". .*Limnology and Oceanography*. DOI:10.1002/lno.11476

Turner, J. N., N. Holmes, S. R. Davis, M. J. Leng, C. Langdon and R. G. Scaife (2015). "A
multiproxy (micro-XRF, pollen, chironomid and stable isotope) lake sediment record for the
Lateglacial to Holocene transition from Thomastown Bog, Ireland." *Journal of Quaternary*
*Science* **30**(6): 514-528.

Van Breemen, N. (1988). Redox Processes of Iron and Sulfur Involved in the Formation of
Acid Sulfate Soils. *Iron in Soils and Clay Minerals*. J. W. Stucki, B. A. Goodman and U.
Schwertmann. Dordrecht, Springer Netherlands**:** 825-841.

Van Der Sleen, P., P. Groenendijk, M. Vlam, N. P. Anten, A. Boom, F. Bongers, T. L. Pons,
G. Terburg and P. A. Zuidema (2015). "No growth stimulation of tropical trees by 150 years
of CO 2 fertilization but water-use efficiency increased." *Nature geoscience* **8**(1): 24.

Verheyden, A., F. De Ridder, N. Schmitz, H. Beeckman and N. Koedam (2005). "High-
resolution time series of vessel density in Kenyan mangrove trees reveal a link with climate."
*New Phytol.* **167**(2): 425-435.

Widlansky, M. J., A. Timmermann and W. Cai (2015). "Future extreme sea level seesaws in
the tropical Pacific." *Science Advances* **1**(8): e1500560.

Zhu, Q. and R. C. Aller (2012). "Two-dimensional dissolved ferrous iron distributions in
marine sediments as revealed by a novel planar optical sensor." *Mar. Chem.* **136**: 14-23.

Zhu, Q., R. C. Aller and Y. Fan (2006). "Two-dimensional pH distributions and dynamics in
bioturbated marine sediments." *Geochim. Cosmochim. Acta* **70**(19): 4933-4949.

Appendix 1.

[Figure]

**Appendix 1. Normalized Mn relative concentrations in mangrove wood over time in living (green dots)**
**and dead (red dots) from upper, mid and lower intertidal areas of mangroves of the Gulf of Carpentaria,**
**Australia. Grey areas indicate the dieback event.**

[Figure]

**Appendix 2. Normalized Ca relative concentrations in mangrove wood over time in living (green dots) and dead (red dots) from upper, mid and lower intertidal areas of mangroves of the Gulf of Carpentaria, Australia. Grey areas indicate the dieback event.**

[Figure]

**Appendix 3.** Cross correlation function (CCF) analysis of the relationship between wood density and
climate data over time at one month resolution over a 12 month period prior and post dieback. Wood
samples are from the upper, mid and lower intertidal zones of the dead (red) and living (green) mangrove
areas. Blue horizontal dashed lines indicate $P< 0.01$ with $n=125$. Grey dashed vertical lines at zero lag
indicate dieback period.